# Abemaciclib is a potent inhibitor of DYRK1A and HIP kinases involved in transcriptional regulation

Ines H. Kaltheuner[1], Kanchan Anand[1], Jonas Moecking[1], Robert Düster [1], Jinhua Wang [2,3], Nathanael S. Gray [4] & Matthias Geyer [1✉]

Homeodomain-interacting protein kinases (HIPKs) belong to the CMGC kinase family and are closely related to dual-specificity tyrosine phosphorylation-regulated kinases (DYRKs). HIPKs are regulators of various signaling pathways and involved in the pathology of cancer, chronic fibrosis, diabetes, and multiple neurodegenerative diseases. Here, we report the crystal structure of HIPK3 in its apo form at 2.5 Å resolution. Recombinant HIPKs and DYRK1A are auto-activated and phosphorylate the negative elongation factor SPT5, the transcription factor c-Myc, and the C-terminal domain of RNA polymerase II, suggesting a direct function in transcriptional regulation. Based on a database search, we identified abemaciclib, an FDA-approved Cdk4/Cdk6 inhibitor used for the treatment of metastatic breast cancer, as potent inhibitor of HIPK2, HIPK3, and DYRK1A. We determined the crystal structures of HIPK3 and DYRK1A bound to abemaciclib, showing a similar binding mode to the hinge region of the kinase as observed for Cdk6. Remarkably, DYRK1A is inhibited by abemaciclib to the same extent as Cdk4/Cdk6 in vitro, raising the question of whether targeting of DYRK1A contributes to the transcriptional inhibition and therapeutic activity of abemaciclib.

[1] Institute of Structural Biology, University of Bonn, Bonn, Germany. [2] Department of Cancer Biology, Dana-Farber Cancer Institute, Boston, MA, USA. [3] Department of Biological Chemistry and Molecular Pharmacology, Harvard Medical School, Boston, MA, USA. [4] Department of Chemical and Systems Biology, Chem-H and the Stanford Cancer Institute, Stanford School of Medicine, Stanford University, Stanford, CA, USA. ✉email: matthias.geyer@uni-bonn.de

Homeodomain-interacting protein kinases (HIPKs) are an evolutionarily conserved kinase family comprising the members HIPK1, HIPK2, and HIPK3 in vertebrates, and, additionally, HIPK4 in mammals. HIPKs belong to the CMGC group of serine/threonine kinases and are part of the dual-specificity tyrosine phosphorylation-regulated kinase (DYRK) family. The DYRK family has been characterized in organisms from yeast to humans and can be divided into five subfamilies, class I DYRKs, class II DYRKs, YAKs, HIPKs, and PRP4s[1].

DYRKs are named after their characteristic dual-specificity, as they auto-phosphorylate a conserved tyrosine in their activation loop, but phosphorylate substrates on serine and threonine residues[2]. For the class II DYRK ortholog from *D. melanogaster* it was demonstrated that the critical tyrosine is *cis*-auto-phosphorylated by the nascent kinase in a transitory intermediate state during maturation at the ribosome[3]. The phosphorylation of the critical tyrosine was later confirmed for members of all DYRK subfamilies by biochemical and structural analyses[4–7]. Also, for all four HIPK members the phosphorylation of the conserved tyrosine has been demonstrated in vivo[8].

In accordance with the close relationship between HIPKs and other DYRK subfamily members, DYRK1A, a class I DYRK member, shares a sequence identity with HIPK3 of 38% within the catalytic domain. Within the HIPK family, HIPK1 and HIPK2 are the most closely related members, sharing about 93% sequence identity in their kinase domains, while HIPK3 is slightly less conserved with 87% identity. The most divergent member of the family is HIPK4, which shares only 50% sequence identity with the catalytic domains of the three other HIPKs. Whereas the smaller and more unique HIPK4 comprises only the kinase domain and potentially unstructured regions[9,10], HIPK1–3 share a complex domain architecture, composed of multiple domains involved in protein–protein interactions. The highly conserved kinase domain is located N-terminal in HIPK1–3, followed by the name-giving homeoprotein-interaction domain (HID) that mediates the interaction with homeodomain transcription factors[11]. C-terminally adjacent to the HID follows a proline, glutamate, serine, and threonine (PEST)-rich domain, mediating proteasomal degradation of these kinases. In HIPK2, a speckle-retention signal (846–941), which is crucial for the localization of HIPK2 to nuclear speckles, overlaps with the PEST domain[12]. Moreover, a C-terminally adjacent autoinhibitory domain (AID) (935–1050) was identified in HIPK2, based on the observation that its removal increases phosphorylation activity[13]. Finally, the C-terminus of HIPK1–3 comprises a region rich in serine, glutamine, and alanine (SQA) residues, which is involved in the interaction with different co-factors[14].

As a consequence of the activation loop autophosphorylation, DYRKs and HIPKs exist in a constitutive, at least partially active state, making further regulatory mechanisms for their directed function in cells indispensable. Indeed, HIPKs are subject to extensive posttranslational modifications (PTMs) such as phosphorylation, acetylation, ubiquitination and SUMOylation, as well as Caspase cleavage[15]. These PTMs are thought to alter the proteins' stability and subcellular localization thereby ultimately controlling kinase activity. Under basal conditions, HIPK1 and HIPK4 are mostly cytoplasmic, whereas HIPK3 is found in the nucleus and HIPK2 undergoes dynamic nucleocytoplasmic shuttling[9,16]. In recent years, the regulatory functions of miRNAs towards protein levels of HIPKs gained increasing attention as circRNAs play important roles in the control of HIPKs gene expression[17].

Several studies have shown that HIPKs phosphorylate a variety of transcription regulators and chromatin modifiers in a remarkable range of diverse signaling pathways[18]. Accordingly, the major function of HIPKs seems to lie in their ability to promote or repress gene expression. Importantly, HIPKs have never been identified as key components of directional downstreaming pathways as it is classically known for Mitogen-activated protein kinases. Instead, HIPKs have been described as "fine tuners", integrating multiple incoming signals into downstream effector pathways[18]. Consequently, HIPKs are involved in the pathology of cancer, chronic fibrosis, and certain neurodegenerative diseases such as Alzheimer´s and Huntington´s disease. Thus, the specific inhibition of HIPKs may be of therapeutic value for multiple diseases.

In this study, we purified the kinase domains of all four HIPKs as recombinant proteins from *E. coli* and *Sf9* cells, characterized their catalytic activity, and determined the crystal structure of the HIPK3 kinase domain at 2.5 Å resolution. Using recombinant protein, we show that HIPKs phosphorylate general components of the transcription machinery, such as c-Myc, SPT5, and the RNA pol II C-terminal domain (CTD). Based on a database screen for potential selective HIPK inhibitors, 15 ATP-competitive small-molecule compounds were selected and tested for their ability to inhibit HIPK3, resulting in the identification of abemaciclib, which was approved by the FDA for the treatment of metastatic breast cancer, as a potent inhibitor of HIPK2, HIPK3, and DYRK1A. Remarkably, $IC_{50}$ values and dissociation constants reveal the inhibition of DYRK1A in the same molecular range as for Cdk4/CycD3 and Cdk6/CycD3. Finally, we determined the crystal structures of HIPK3 and DYRK1A bound to abemaciclib at 2.8 and 1.8 Å resolution, enabling the structure-guided optimization for selective HIPKs inhibitors. Our findings establish HIPKs as direct transcription regulating kinases which may already be serendipitously targeted by the established drug abemaciclib.

## Results

**Crystal structure of the human HIPK3 kinase domain.** To determine the structure of the human HIPK3 kinase domain, residues 159–562 were expressed in baculovirus infected *Sf9* insect cells. This construct extends N- and C-terminally beyond the canonical kinase domain, but showed high solubility (Fig. 1a, b). Recombinant protein was displayed to homogeneity using affinity purification and size exclusion chromatography (SEC). Crystals were grown in the presence of ADP and $MgCl_2$ by the hanging drop vapor diffusion method. The structure was determined to a resolution of 2.5 Å and refined to an $R_{work}$ of 0.24 and $R_{free}$ of 0.27 with excellent stereochemistry (Supplementary Table 1). One HIPK3 monomer forms the asymmetric unit cell of the protein crystal. Despite its presence in the crystallization condition, no crystallographic density was seen for ADP or magnesium, implying that the kinase is in the nucleotide free apo-state.

In the structure, a continuous polypeptide chain from residue 184 to 550 was determined for HIPK3, exhibiting the classical kinase fold as seen, e.g., for Cdk2[19] with an N-terminal lobe (197–278) and a C-terminal lobe (279-526) including a fully ordered activation segment (Fig. 1c, d). The structure of HIPK3 exhibits an RMSD value of 0.58 Å over 290 Cα atoms at a sequence identity of 88% compared to the human HIPK2 crystal structure[20]. Significant deviations occur in the activation loop, the CMGC-specific insert region, and the loop around position 480 connecting helices αN with αH. At the N-terminus, 13 additional residues preceding the canonical kinase domain are seen in the electron density map, forming an extended stretch in an antiparallel conformation to the β1 strand of the N-lobe. Although these residues adopt a similar conformation as the DH box in DYRK1A and DYRK2, the two central tyrosines that stabilize the association of the DH box to the kinase domain in DYRKs[4], are not conserved in HIPKs (Fig. 2). Another 24

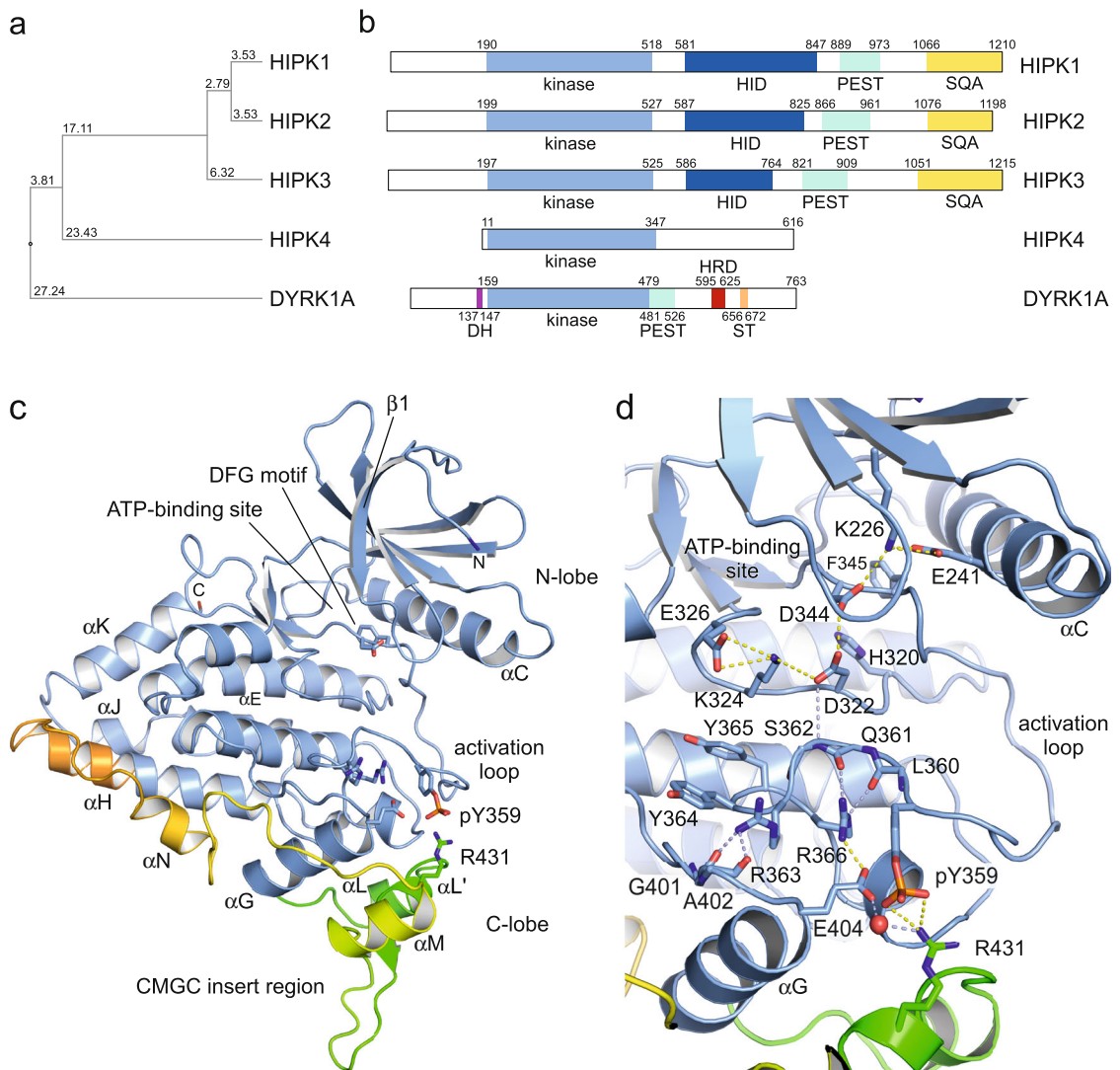

**Fig. 1 Structure of the human HIPK3 kinase domain. a** Phylogenetic tree of human HIPK1–4 and DYRK1A kinases illustrating the homology of the catalytic domains. The average distance tree was calculated by percentage identity using Jalview[63]. **b** Domain architecture of human HIPK1–4 and DYRK1A. HID homeoprotein-interacting domain, PEST proline, glutamate, serine, threonine-rich region, SQA serine, glutamine, alanine-rich region, DH DYRK homology box, HRD histidine-rich domain, ST serine/threonine-rich region. **c** Crystal structure of the human HIPK3 kinase domain (PDB: 7O7I). Key elements as the αC helix, the DFG motif, and the phosphorylated tyrosine within the activation loop are indicated. The CMGC insert region (residues 416–493) embedded between canonical helices αG and αH is colored from green to orange. **d** Coordination of the active center of the HIPK3 kinase. Electrostatic and hydrogen-bond interactions between the phosphorylated tyrosine of the activation loop, pY359, the RYYR element, the HAD motif with the general base, the DFG motif, the catalytic lysine K226 and the coordinating glutamate E241 of the αC helix are shown.

residues are resolved at the C-terminus extending the chain to position 550 by two helices (αJ and αK) that form an α-barrel with helices αI and αE of the C-lobe at the back of the ATP-binding site. Despite being in the apo state, the HIPK3 structure adopts the active kinase conformation with a phosphorylated tyrosine in the activation loop, the DFG motif and the αC helix in the "in"-position, and a fully accessible ATP-binding site (Fig. 1d).

**HIPK3-specific structural features**. The activation loop of DYRKs contains a YxY element, whose second tyrosine is auto-phosphorylated for kinase activation[4]. This motif is altered to STY in HIPK1–3 and EPY in HIPK4 (Fig. 2). In our crystal structure of HIPK3, the conserved tyrosine Y359 constitutes the only phosphorylated residue. The activating tyrosine is neighboring the "CMGC arginine" R366, which is part of the $R_{363}$YYR

motif and a characteristic feature of the CMGC kinase family[21]. In all known protein structures of DYRKs and of HIPK2, the phosphate group of the conserved pY forms strong electrostatic interactions with the guanidine side chains of these two arginines, stabilizing the activation loop in the kinase active conformation (Fig. 3a). For HIPK3 instead, we find that the phosphate group of pY359 forms a 3.0 Å salt bridge with the side chain of R431, which is part of the HIPK specific insert region (Supplementary Fig. 1a, b). The RYYR motif remains at the same position as in DYRKs and HIPK2, yet, R363 is hydrogen-bonded to the car-bonyl group of G401 and R366 forms a tight salt bridge (2.6 Å) to E404 (Fig. 1d). This interaction develops into an electrostatic network with a water-mediated hydrogen bond to R431. In addition, the CMGC arginine R366 forms hydrogen bonds to the main-chain oxygen of L360 and Q361, stabilizing the turn in this strained backbone conformation of the activation loop. The unexpected conformation of the pY359 phosphate group toward

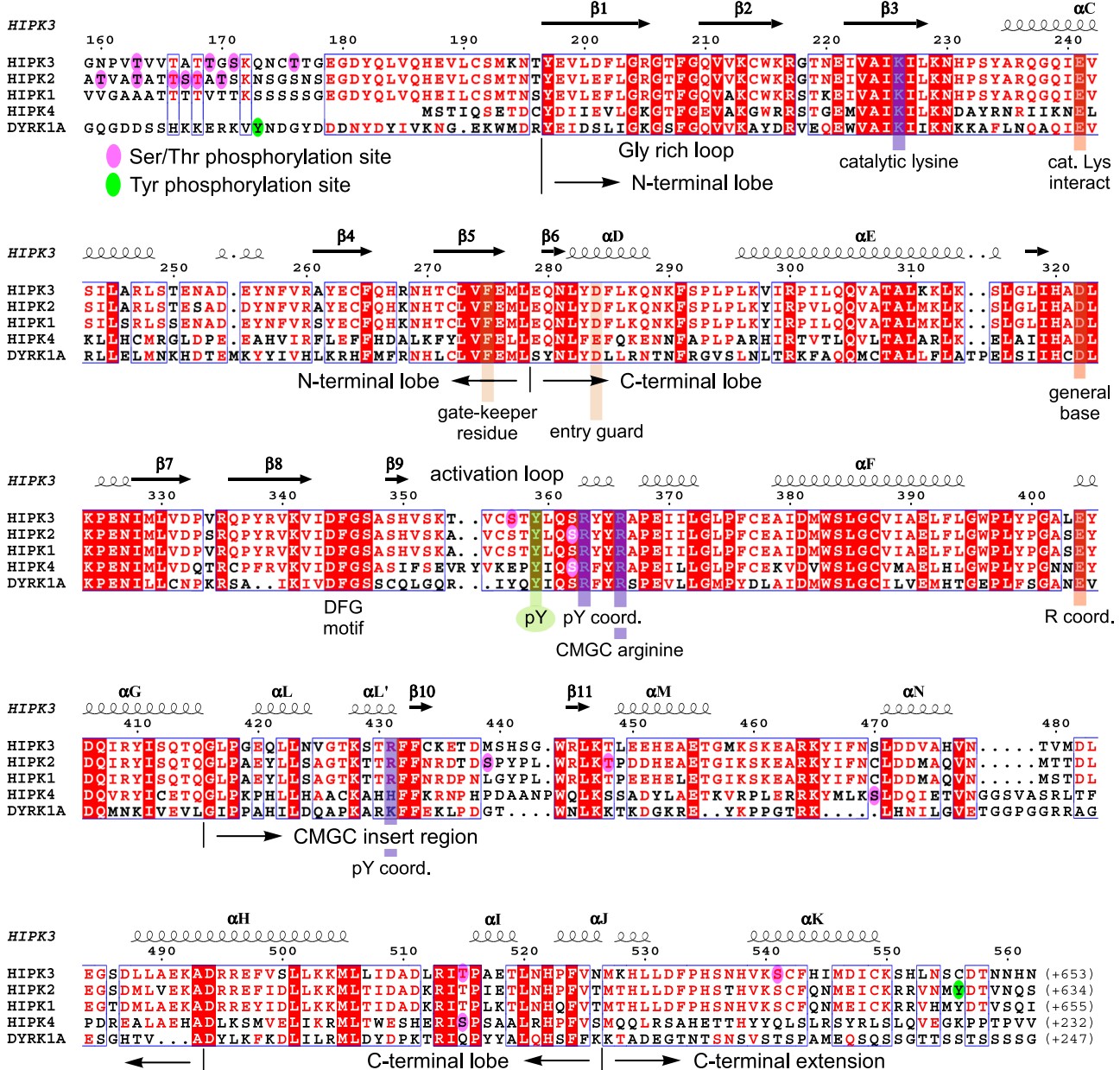

**Fig. 2 Sequence alignment of human HIPK1–4 and DYRK1A kinase domains.** Sequence alignment of the kinase domain of human HIPKs and DYRK1A over the entire length of the crystallized HIPK3 construct (159–562). Secondary structure elements are indicated for HIPK3 as determined for the apo-HIPK3 structure. Characteristic sequence motifs, including the phosphorylated T-loop tyrosine, and functional regions are indicated. Serine/threonine or tyrosine residues found to be phosphorylated are circled magenta or green, respectively. Residues conserved in all kinases are boxed red, and those that are similar have red characters. The sequence alignment was prepared with MultAlin. The secondary structure alignment was prepared with ESPript. UniProt accession numbers are: Q86Z02 (HIPK1), Q9H2X6 (HIPK2), Q9H422 (HIPK3), Q8NE63 (HIPK4), and Q13627 (DYRK1A).

R431 is accompanied by an intermolecular salt bridge from a symmetry related molecule (Supplementary Fig. 2a). D482 at the tip of the loop between helices αN and αH of a neighboring molecule is taking the place, forming tight salt bridges to R363 and R366 of the RYYR motif. A weak intermolecular salt bridge is additionally formed between the pY359 phosphate group and K297 of the symmetry related molecule. The electrostatic surface potential surrounding pY359 in HIPK3 is shown in Supplementary Fig. 2b, c.

The CMGC insert region is characteristic for this kinase branch, serving as a specific binding platform for signaling partners[21]. With 78 to 84 residues inserted in the C-lobe between canonical helixes αG and αH (colored green to orange in Fig. 1c),

HIPKs possess the longest insert region of all CMGC kinases. In HIPK3, the insert region is built by four helices with two short antiparallel β-strands and extends across the base of the C-lobe leading into a striking extension of helix αH. Following two short helices (αL and αL′), the β-hairpin (433–446) reaches out by a loop of eight residues before turning toward helix αG and meandering over 40 residues with two long loops and two helices (αM and αN) into the canonical C-lobe. Interestingly, this conformation varies significantly for the known structures of HIPK2, DYRK1A, DYRK2, and DYRK3, underlining the diversity and specificity gained by the CMGC insert region (Fig. 3b). For example, a phosphorylation site in HIPK2 (pS441) bends the β-hairpin loop toward helix αL, while this loop is three residues

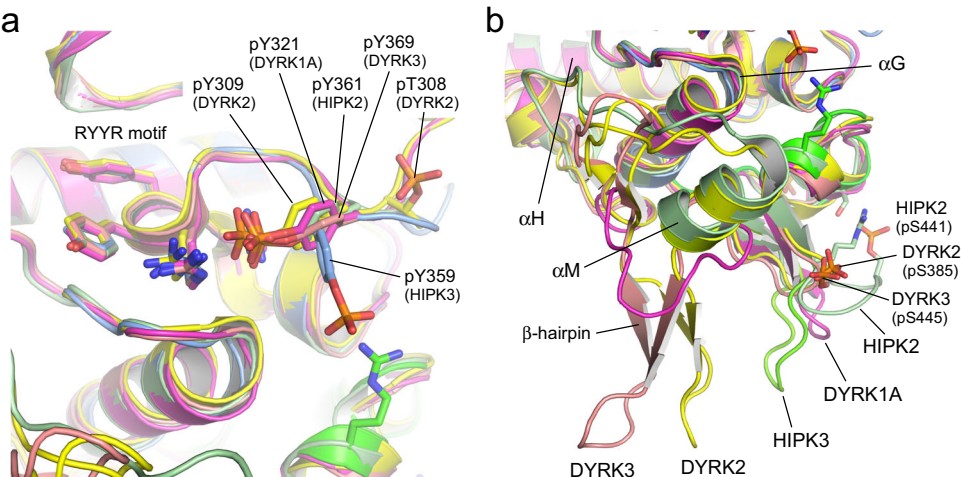

**Fig. 3 Coordination of the phosphorylated tyrosine and the CMGC insert region in HIPKs and DYRKs. a** In HIPK2, DYRK1A, DYRK2, and DYRK3 the phosphorylated tyrosine residue of the activation loop forms salt bridges with the arginines of the R(Y/F)YR motif. In HIPK3 instead, pY359 interacts with R431 of the CMGC insert region. **b** Variations of the β-hairpin loop conformation in the CMGC insert region. Protein structures shown in this figure are HIPK2 (6P5S, palegreen), HIPK3 (7O7I, blue/green), DYRK1A (2VX3, magenta), DYRK2 (3K2L, yellow), and DYRK3 (5Y86, salmon).

shorter in DYRK1A (Supplementary Fig. 1c). In DYRK2 and DYRK3 instead, the β-hairpin loop is shifted towards the position of helix αM in HIPK3 (Fig. 3b). Beside these divergent conformations, DYRK2 contains two and DYRK3 one phosphorylation site in the insert region, contributing further to the specificity of this characteristic region.

**Autophosphorylation and activity of HIPKs.** It is not completely understood to date, whether autophosphorylation at the conserved tyrosine in the activation loop is sufficient for HIPKs activation or if phosphorylation at other residues, either by autophosphorylation or by other kinases, are additionally required[6,8,15]. We prepared recombinant HIPK1–4 from *Sf9* insect cells to gain native phospho-modifications and prepared HIPK2-3 additionally from *E. coli* bacterial cells, which was not successful for HIPK1 and HIPK4 due to protein instability. We were curious, whether post-translationally modified HIP kinases from *Sf9* cells would show differences in their activity compared to auto-phosphorylated kinases from *E. coli*. Thus, we tested enzymatic activity in in vitro kinase assays using [$^{32}$P]-γ-ATP and c-Myc as a substrate (Fig. 4a). Interestingly, all kinases were active and showed no differences in activity independent of the expression system, indicating that no additional phospho-modifications apart from autophosphorylation are required for general HIP kinase activation. To identify modified phospho-sites and the absolute degree of protein phosphorylation, we used peptide mass fingerprint analysis and determined the intact masses of recombinant kinases using ESI LC-MS (Supplementary Table 2 and Fig. 4b). Interestingly, proteins expressed in *E. coli* showed more phospho-residues on average compared to insect cell derived proteins, which we assume results from a higher degree of posttranslational modification regulation in insect cells. All HIP kinases contained at least one phosphorylation and the activation loop tyrosine was always phosphorylated to a high extent. The total mass of HIPK2 from *E. coli* showed indeed up to nine phosphorylations. Particularly Ser/Thr phosphorylations in the N-terminal region preceding the kinase domain were identified for HIPK2 and HIPK3, which seems reminiscent to DH domain phosphorylations in DYRK1A and DYRK2[4]. Importantly, a second phosphorylation site in the activation loop as, e.g., the sites pTpY in DYRK2 or pTEpY in ERK2, was only seen at a low coverage for pS357 in HIPK3 (making a pSTpY motif)

and for the serine three residues downstream of the pY site in HIPK2 and HIPK4 (Supplementary Table 2). This latter modification results in a pYxxpSRYYR signal, those double phosphorylation was found to be essential for HIPK2 kinase function[6]. For a visualization, all phosphorylation sites found in our proteomic analyses of HIP kinases were mapped onto the structure of HIPK3 (Supplementary Fig. 3).

**HIPKs phosphorylate general transcription factors and RNA pol II CTD in vitro.** In metazoans, gene transcription and co-transcriptional RNA processing are tightly regulated by a complex interplay between RNA pol II and a variety of general transcription factors. The C-terminal domain (CTD) of the Rpb1 subunit of human RNA pol II consists of 52 hepta-peptide repeats with the consensus sequence $Y_1S_2P_3T_4S_5P_6S_7$, harboring multiple sites for posttranslational modifications[22]. After transcription initiation, RNA pol II is stalled 20-50 bases downstream of the transcription start site, which is mediated by Negative Elongation Factor and DRB sensitivity inducing factor (DSIF) that is composed of SPT4/SPT5[23,24]. To release RNA pol II from promoter-proximal pausing into productive elongation, positive transcription elongation factor b (P-TEFb), also known as Cdk9/Cyclin T1, phosphorylates RNA pol II CTD at Ser2 and Ser5 residues, as well as the C-terminal region of DSIF component SPT5[25]. Similar to P-TEFb, DYRK1A has been shown to phosphorylate RNA pol II CTD at Ser2 and Ser5 residues at crucial genes for development and tissue homeostasis[26,27]. Considering the provocative similarities between the DYRK and HIPK families, we investigated whether HIPKs are involved in the direct regulation of the transcription machinery. We tested all four HIPKs, DYRK1A, and P-TEFb in in vitro kinase assays using [$^{32}$P]-γ-ATP and different recombinant substrate proteins (Fig. 5a). Interestingly, under the in vitro conditions employed, we found that the kinase activity of DYRK1A as well as HIPK1–3 towards the C-terminal region of SPT5 is even more pronounced than of P-TEFb. Likewise, all four HIPKs showed robust kinase activity toward the transcription factor and proto-oncogene c-Myc, as well as the human full-length GST-CTD$_{[52]}$.

To define the specificity of HIPKs towards the precise phospho-residues on the CTD heptad repeats, we performed immunoblotting analysis using monoclonal antibodies raised against pTyr1, pSer2, pThr4, pSer5, and pSer7 marks (Fig. 5b).

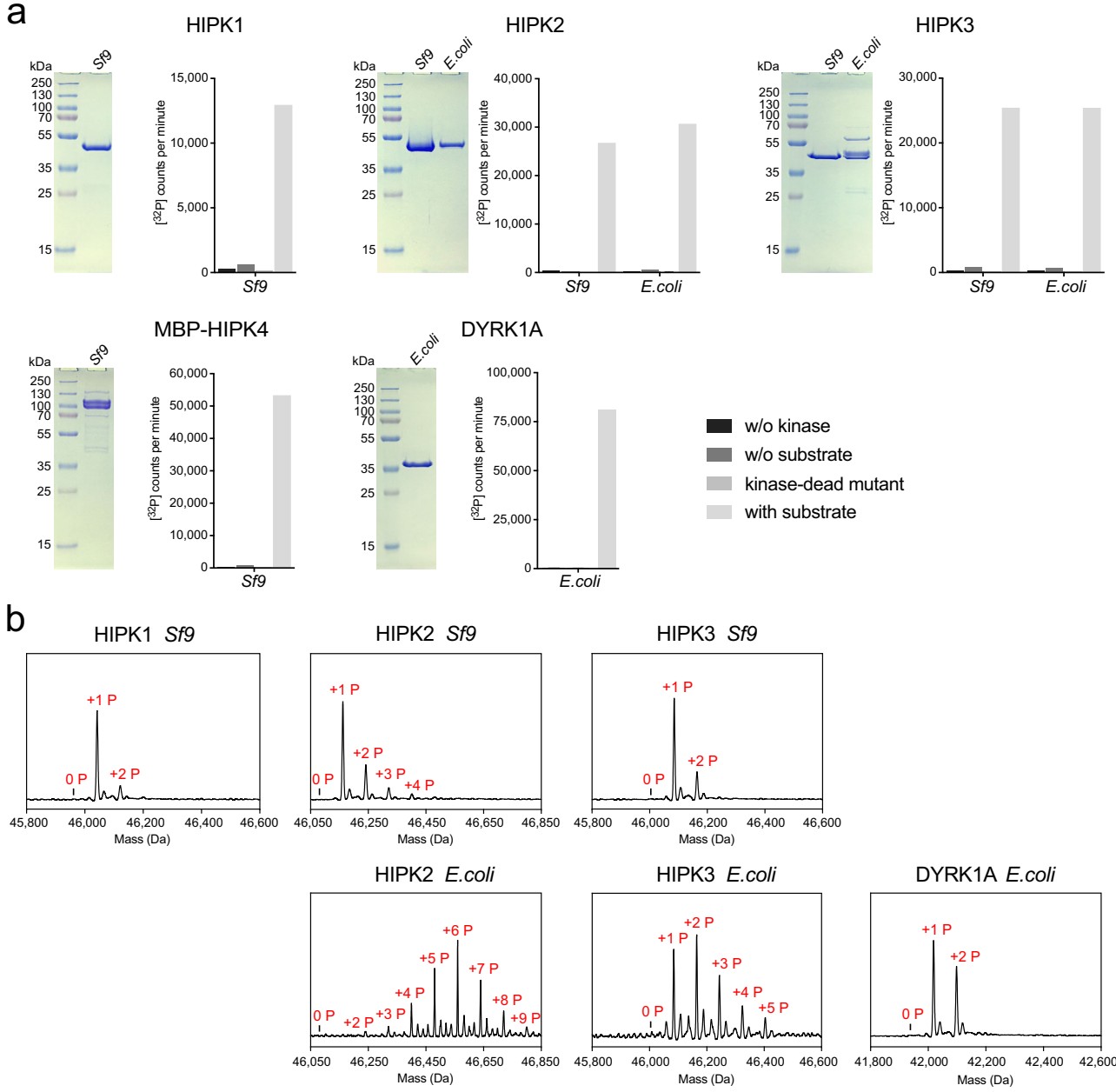

**Fig. 4 Activity and phosphorylation status of recombinant HIP kinases. a** Purification and activity measurements for all four HIPKs and DYRK1A. Recombinant protein kinases were either purified from *Sf9* insect cells or from *E. coli* bacterial cells and analyzed by SDS-PAGE. Kinase activity was assessed using in vitro kinase assays with 0.2 mM [$^{32}$P]-$\gamma$-ATP, which was incubated for 30 min either without kinase, without substrate, as a kinase-dead mutation (HIPK1$_{D315N}$, HIPK2$_{D324N}$, HIPK3$_{D322N}$, and HIPK4$_{D136N}$) or as wild-type kinase in the presence of 0.2 µM kinase and 10 µM His-c-Myc as a substrate. Measurements were performed as duplicates ($n = 2$ biologically independent samples) and are depicted as mean. **b** Molecular masses of intact kinases determined by ESI-(LC)-MS indicate the total number of phosphorylations. Source data are provided as a Source Data file.

Similar to DYRK1A, HIPK1–3 exhibit phosphorylation preferences for Ser2 and Ser5 residues. Phosphorylation of Tyr1 and Thr4 residues instead was not detected. Remarkably, HIPK4 phosphorylates the pol II CTD at Ser7 residues, highlighting the different character of this kinase in comparison to its family members. Consistently, in time-course experiments with gel shift assays (Fig. 5c, d) the migration of the GST-CTD[52] band from a hypo- (IIa) to a hyper-phosphorylated (IIo) state was clearly detectable for all four HIPKs and DYRK1A.

In previous studies it has been shown that pre-phosphorylation of a specific CTD residue at Tyr1 or Ser7 primes RNA pol II CTD recognition by P-TEFb, Cdk12, and Cdk13[28–31]. We therefore analyzed HIPKs and DYRK1A activity towards a set of synthetic CTD peptides, with each peptide comprising three hepta-repeats followed by a polyethylene linker and two arginines (Fig. 5e). Additional to the CTD consensus sequence, CTD peptides were uniformly phosphorylated at position Tyr1, Ser2, Thr4, Ser5, Ser7, or with lysines at position 7, the most frequent alteration in the human CTD consensus repeats[22]. Even though background signals were high for HIPK2 and HIPK3, a similar trend could be detected for all five kinases, exhibiting the highest phosphorylation activity towards consensus CTD and no activity above background for Ser2, Ser5, and Ser7 modified peptides. Lys7 substituted peptides were recognized as substrates nearly

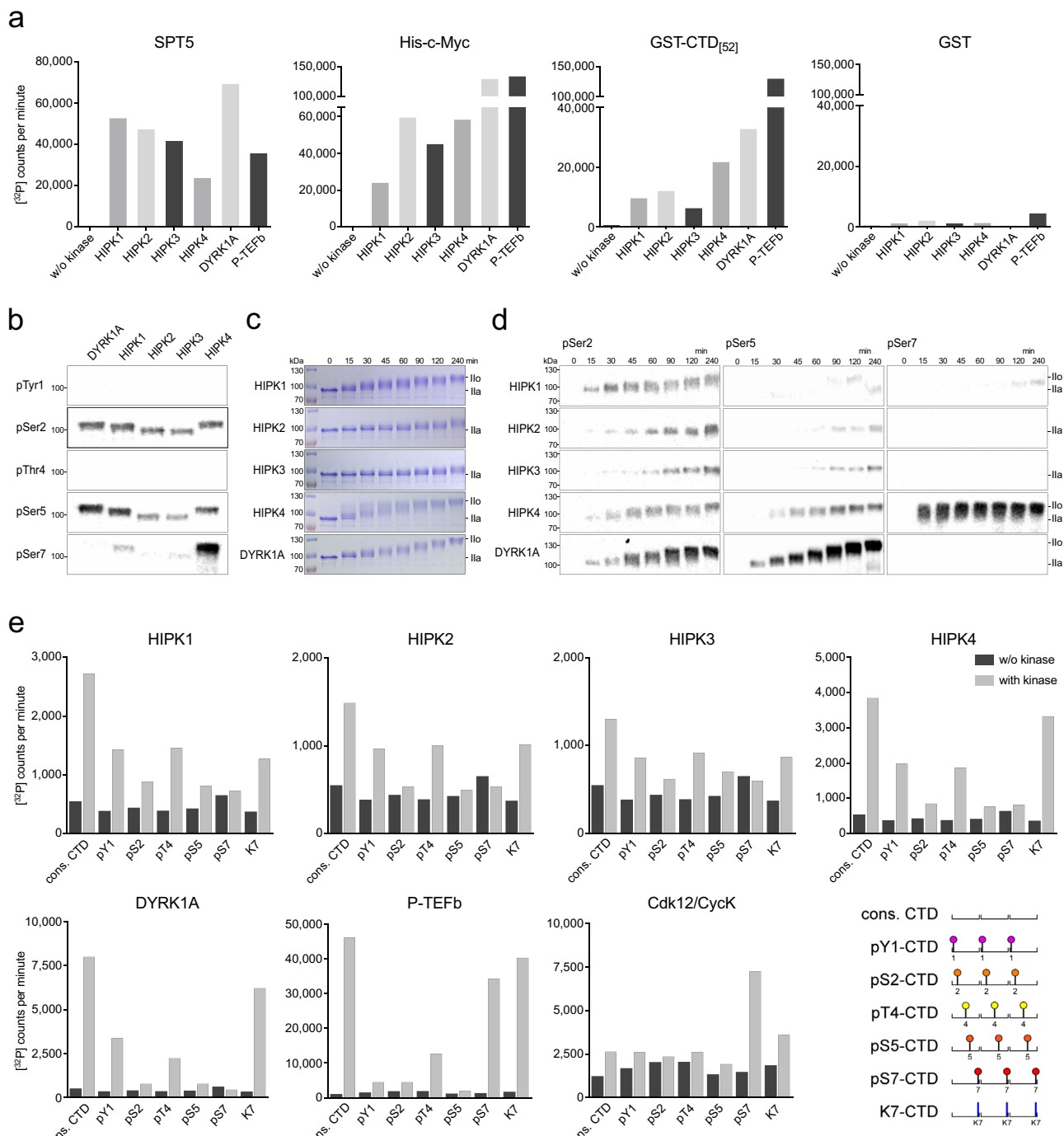

**Fig. 5 HIPKs phosphorylate general transcription factors and RNA pol II CTD. a** For radioactive kinase activity assays 10 μM human SPT5, His-c-Myc, GST-CTD[52], and GST as a control were incubated with 0.2 mM [32P]-γ-ATP either with or without 0.2 μM kinase for 30 min. **b** For in vitro kinase assays 10 μM GST-CTD[52], 0.2 mM ATP, and 0.2 μM kinase were incubated for 120 min, and visualized by immunoblotting using RNA pol II CTD specific phosphoantibodies pTyr1 (3D12), pSer2 (3E10), pThr4 (6D7), pSer5 (3E8), and pSer7 (4E12). **c** Gel shift assays were performed as in (**b**) but in time-course series and visualized by Coomassie staining in SDS gels. Upon phosphorylation the GST-CTD[52] band shifts from the IIa to the IIo form. **d** In vitro kinase assays were performed as in (**b**) but as time-course series, visualized by immunoblotting. **e** Radioactive kinase assays were performed as in (**a**) using 150 μM pre-phosphorylated CTD Peptides as a substrate. Each CTD peptide contained three consensus hepta-repeats with either no modification (cons. CTD) or with phosphorylation marks continuously set at one residue of the heptad sequence as depicted in the cartoon. The principle was used for Tyr1, Ser2, Thr4, Ser5, Ser7 or a peptide that contained a lysine at position 7. Measurements were performed as duplicates (*n* = 2 biologically independent samples) and are depicted as mean. Source data are provided as a Source Data file.

to the same extent as consensus CTD by HIPK4 and DYRK1A and with moderate activity by HIPK1, HIPK2, and HIPK3, followed by Tyr1 and Thr4 modified CTD peptides. However, in contrast to CDKs 9, 12, and 13, a preference of S7 prephosphorylated CTD substrates over consensus CTD was not observed for HIPKs. Together, these data support the intriguing possibility that DYRK1A as well as the HIPK family may have essential functions as transcription regulation kinases, highlighted by similar substrate phosphorylation patterns of SPT5 and RNA pol II CTD as it has been known for P-TEFb.

**Abemaciclib is a potent inhibitor of HIPK2, HIPK3, and DYRK1A.** HIPK3 has been implicated in the pathology of different diseases such as prostate cancer[32], type II diabetes[33], and Huntington's disease[34,35]. However, only a few protein kinase inhibitors have been described to date for the specific inhibition of HIPKs[20,36–38]. To identify potent small-molecule inhibitors of HIPK3, we performed a database search based on a kinome-wide binding assay[39] of over a thousand compounds. Six inhibitors were selected from the compound library of the Gray laboratory (XMD8-70, XMD8-62-i, JWD-065, HTH-01-091, CVM-05-145-3, and CVM-06-033-2), as well as the reported inhibitors dinaciclib (SCH727965), palbociclib (PD0332991), abemaciclib (LY2835219), GSK1059615, and AS601245. In addition, the widely used pan-selective CDK inhibitors flavopiridol, (R)-roscovitine, and staurosporine were chosen as well as the described Cdk9 inhibitor NVP-2. In a first screen, we tested these 15 candidate compounds for their ability to inhibit HIPK3 activity in vitro. Compounds were preincubated at concentrations of 1, 10, and 100 µM with 0.2 µM kinase and 200 µM radioactive ATP for 5 min, followed by the addition of 50 µM His-c-Myc as a substrate to start the reaction and 15 min time span of the kinase reaction (Fig. 6a). Whereas some compounds such as flavopiridol, roscovitine, and dinaciclib showed almost no inhibitory effect on HIPK3, abemaciclib, GSK1059615, JWD-065, HTH-01-091, and CVM-06-033-2 (structures shown in Supplementary Fig. 4) reduced HIPK3 activity most potently. Staurosporine and palbociclib instead showed only modest efficacy toward HIPK3.

To analyze the inhibitory potential of the five best hits towards HIPK3 in more detail, we tested compound concentration series in a range from 3.1 nM to 1 mM at 0.2 µM kinase and 200 µM ATP concentrations and used a sigmoidal fit to determine in vitro $IC_{50}$ values from these dose-response measurements (Fig. 6b). Interestingly, the highest efficacy was achieved with abemaciclib ($IC_{50}$ 467 nM), a Cdk4/Cdk6 inhibitor, which was approved by the FDA in 2017 for the treatment of hormone receptor positive (HR+) and HER2-negative metastatic breast cancer[40–42]. Also, GSK1059615 ($IC_{50}$ 1079 nM) and HTH-01-091 ($IC_{50}$ 1597 nM) showed robust inhibition of HIPK3. We were curious, whether the potency of abemaciclib towards HIPK3 acts in a significant range compared to its described targets Cdk4 and Cdk6. Thus, we determined $IC_{50}$ values for the inhibition of all four HIPKs, DYRK1A, Cdk4/CycD3, and Cdk6/CycD3 using GST-Rb1 as substrate for Cdk4 and Cdk6. In addition, inhibition of Cdk9/CycT1, a suggested off-target of abemaciclib[41,43], was analyzed (Fig. 6c). Strikingly, our in vitro data show that DYRK1A is inhibited by abemaciclib ($IC_{50}$ 93 nM) to the same extent as Cdk4/CycD3 ($IC_{50}$ 78 nM) and Cdk6/CycD3 ($IC_{50}$ 116 nM). Similar to HIPK3, abemaciclib showed robust inhibition of HIPK2 ($IC_{50}$ 668 nM) but was less potent towards HIPK1 ($IC_{50}$ 4.53 µM) and HIPK4 ($IC_{50}$ 10.36 µM). Overall, abemaciclib appears to inhibit kinases from the HIPK and DYRK families of the CMGC kinase group to a significant extent compared to its described targets Cdk4 and Cdk6.

**Abemaciclib binds with similar affinities to HIPK2, HIPK3, and DYRK1A as to Cdk4.** Surface plasmon resonance (SPR) was used to further characterize the binding affinity of abemaciclib towards HIPK2, HIPK3, DYRK1A, and Cdk4/CycD3 (Fig. 7a). Kinases were immobilized by amine-coupling onto a CM5 chip and increasing concentrations of abemaciclib were used for single-cycle kinetic measurements. Intriguingly, DYRK1A and Cdk4/CycD3 show dissociation constants in the low nanomolar range with 8.6 and 1.4 nM, respectively, whereas HIPK2 and HIPK3 exhibit a $K_D$ even in the sub-nanomolar range with 0.8 nM. In addition, thermal stability shift assays were performed for all four HIPKs, DYRK1A, Cdk4/CycD3, and Cdk9/CycT1, assessing the stability of these kinases upon binding abemaciclib at increasing concentrations (Fig. 7b, c). We used 5 µM kinase concentration preincubated with 1, 10 or 100 µM abemaciclib compared to a DMSO control. HIPK1, HIPK2, HIPK3, DYRK1A, and Cdk4/CycD3 were significantly stabilized with melting temperatures shifting up to +8.7 °C for HIPK1 and +7.9 °C for Cdk4/CycD3, followed by +7.2 °C and +7.4 °C for HIPK2 and HIPK3, respectively, and 5.2 °C for DYRK1A upon incubation with 100 µM abemaciclib (Supplementary Table 3). Cdk9/CycT1 and MBP-HIPK4 instead were only moderately stabilized with increasing melting temperatures of +3.3 °C and +1.9 °C, respectively.

**Crystal structures of HIPK3 and DYRK1A bound to abemaciclib.** To understand the interaction of abemaciclib with HIPK3 and DYRK1A in detail we crystallized the complexes and solved the structures to 2.8 and 1.8 Å resolution, respectively. Despite its widespread therapeutic use, only one structure containing abemaciclib is available in the protein data bank to date, which is the Cdk6–abemaciclib crystal structure determined at 2.3 Å resolution (PDB code: 5L2S)[44]. Similar as in Cdk6, abemaciclib interacts intensively via its central 2-aminopyrimidine group with the hinge region of HIPK3 and DYRK1A (Fig. 8a, b). The backbone amino and carbonyl groups of L241 in DYRK1A form hydrogen bonds with the three nitrogen atoms of the aminopyrimidine-pyridine unit (Supplementary Fig. 5a). At the benzimidazole head, the fluorine atom forms H-bonds with the catalytic lysine K188 and the coordinating glutamate E203 of the αC helix in DYRK1A, which is in the 'in'-position. The G-loop is fully ordered and no additional ion, as, e.g., the chloride in the active site of DYRK1A with inhibitor DJM2005[4], is seen in our structure. Towards the rear part of the compound, the pyridine and the piperazine ring form hydrophobic interactions with I165 at the G-loop (Fig. 8b). In addition, the positively charged piperazine ring is stabilized by lying against a solvent-exposed rim consisting of N244 and D247 at the transition between the N- and C-lobe (Fig. 8c). This latter interaction is thought to be one defining element for the reported specificity of abemaciclib for Cdk4/6 compared to Cdk1/2/3/5[44]. In Cdk6, these two residues correspond to D104 and T107, and both are identical in Cdk4 (Supplementary Fig. 5b). In Cdk1/2/3/5 the T107 position of Cdk6 (D247 in DYRK1A) is a lysine, which might cause electrostatic repulsion with the piperazine and thereby lower the potency of abemaciclib for Cdk1/2/3/5 (Fig. 8d). Conversely, in HIPK1–3 and DYRK1A/B this residue is a negatively charged aspartate, and a larger glutamate in HIPK4 and DYRK2-4, possibly allowing electrostatic association (Supplementary Fig. 5b). These two defining residues might explain some of abemaciclib's specificity within the CMGC kinase group. Moreover, two tightly packed citrate molecules were identified in the high-resolution structure of DYRK1A, keeping the active site open like a clamp at the nucleotide exit position (Supplementary Fig. 5c, d). Two negatively charged carboxy groups of one citrate molecule align to

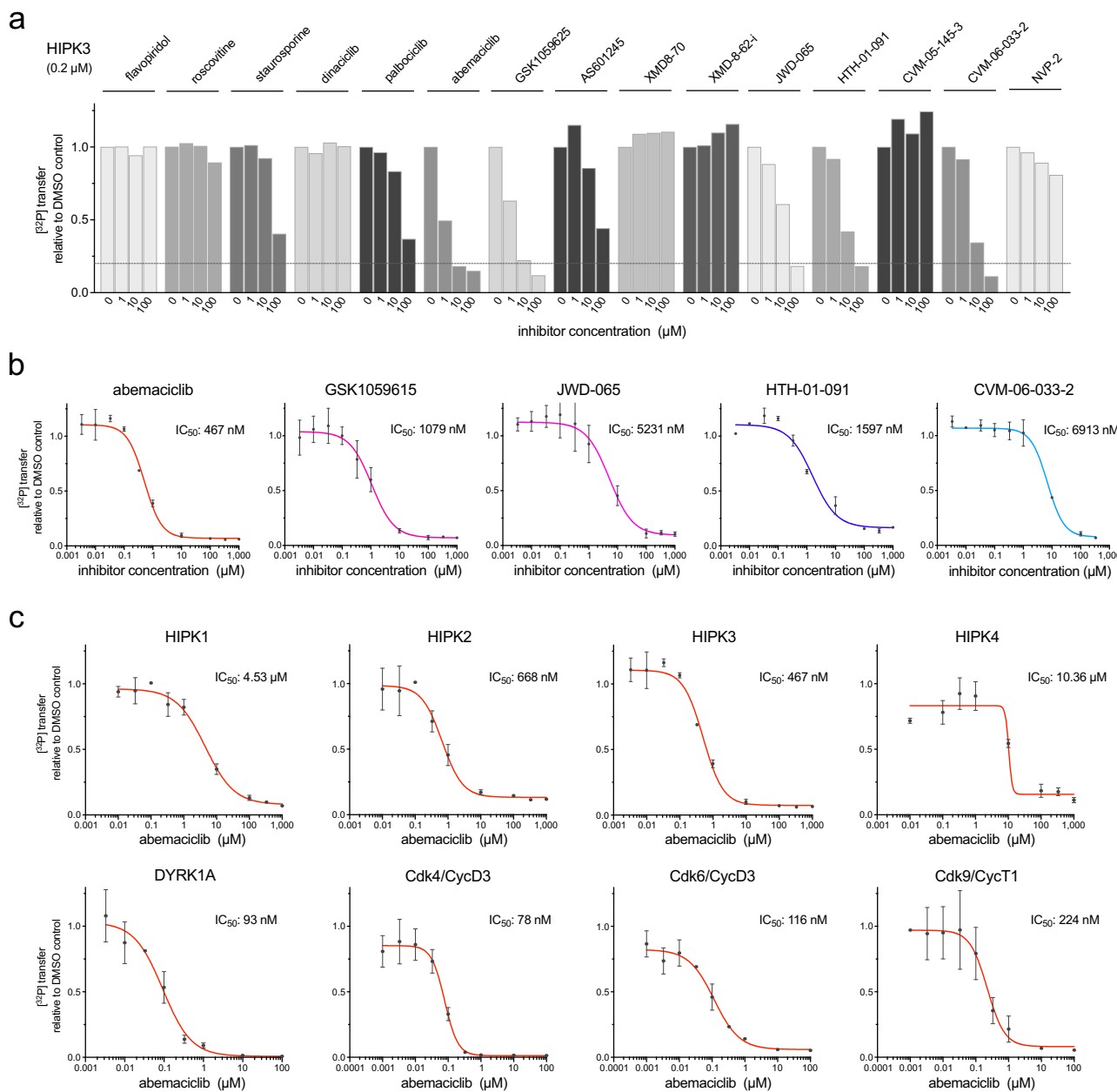

**Fig. 6 Abemaciclib is a potent inhibitor of HIPK2, HIPK3, and DYRK1A. a** A panel of 15 small-molecule inhibitors was tested at 1, 10, and 100 μM concentration for the inhibition of HIPK3 using radioactive kinase activity assays. Compounds were preincubated for 5 min with 0.2 μM kinase and 0.2 mM [$^{32}$P]-γ-ATP, followed by the addition of 10 μM His-c-Myc as a substrate and another incubation for 15 min. Measurements were performed as duplicates (*n* = 2 biologically independent samples) and are depicted as mean. **b** In vitro kinase assays were performed as in (**a**) testing concentration series of the best five small-molecule inhibitors for HIPK3. All data are depicted as mean ± SD from duplicates (*n* = 2 biologically independent samples). A sigmoidal fit was used to calculate IC$_{50}$ values. **c** In vitro kinase assays were performed as in (**a**) using concentration series of abemaciclib for the inhibition of all four HIPKs, DYRK1A, Cdk4/CycD3, Cdk6/CycD3, and Cdk9/CycT1. All data are depicted as mean ± SD from duplicates (*n* = 2 biologically independent samples). Source data are provided as a Source Data file.

the position, where otherwise the γ-phosphate of ATP is located and the transition to the Ser/Thr residue of the substrate takes place.

The interface of HIPK3 to abemaciclib is almost identical compared to DYRK1A, particularly the H-bond formation in the hinge region to the 2-aminopyrimidine and pyridine rings are similar (Supplementary Fig. 6). Yet, one difference to Cdk6 is a hydrophobic methionine in HIPK3 (M277) and DYRK1A (M240) where Cdk6 contains a histidine (H100), whose imidazole ring forms a water-mediated hydrogen bond to the pyridine nitrogen[44] (Fig. 8b, e). Whereas no water molecule is found in a

similar coordination in HIPK3 and DYRK1A, a weak hydrogen-bond could be formed by the methionine sulfur to the nitrogen in the linker between the aminopyrimidine and the pyridine compensating for the water-mediated interaction in Cdk6. Toward the base, the piperazine ring in HIPK3 is in the same saddle conformation as in the other two structures, but is bent upwards, which may be due to L203 at the G-loop, which is a β-branched isoleucine in DYRK1A and Cdk6. Finally, with 536 and 557 Å$^2$ the buried surface areas of abemaciclib in DYRK1A and HIPK3 appear similar to that of Cdk6 with 547 Å$^2$, although the compound seems to be a little deeper soaked into the ATP-binding

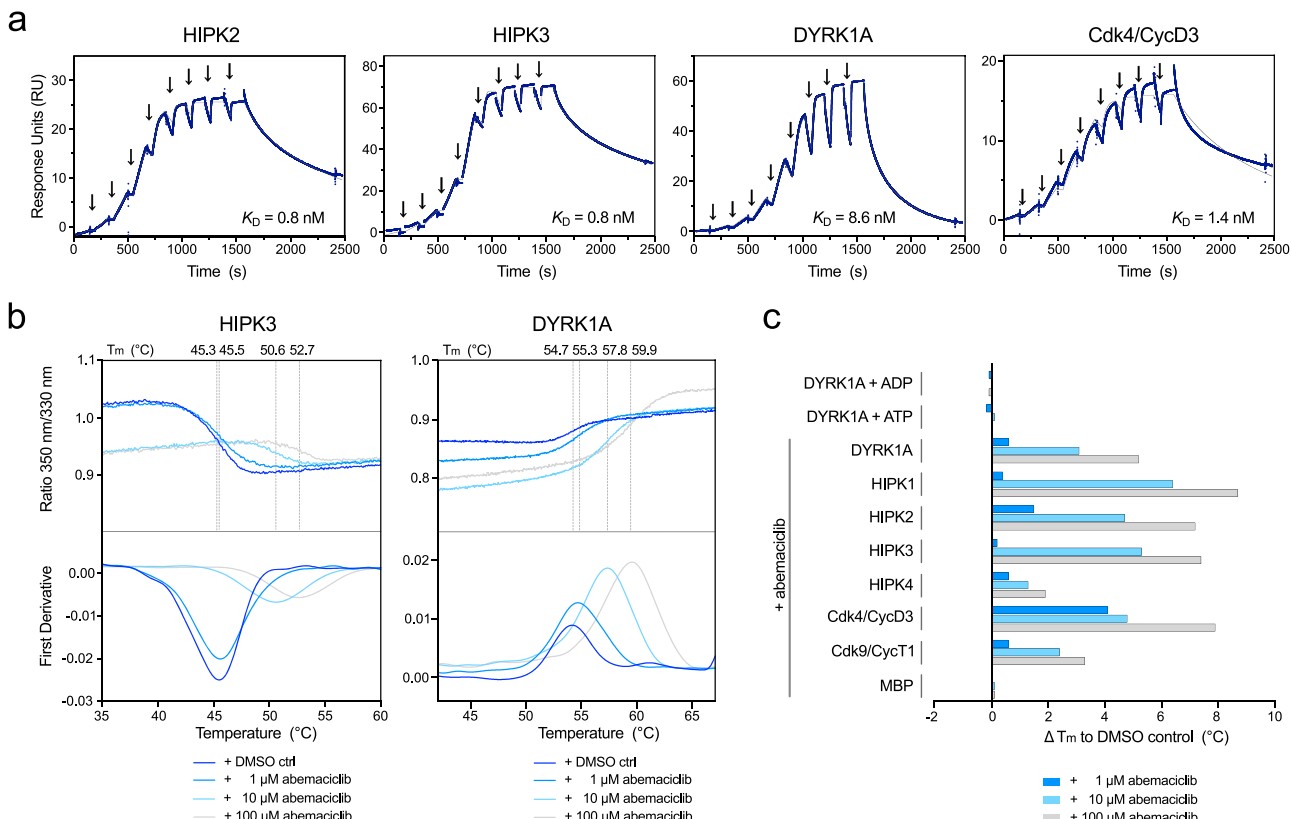

**Fig. 7 Binding and affinity measurements of abemaciclib towards HIPK2, HIPK3, DYRK1A, and Cdk4/CycD3. a** Surface plasmon resonance (SPR) measurements were performed using single-cycle kinetics. Kinases were immobilized to the chip surface by amine coupling. Abemaciclib was injected at increasing concentrations of 0.55, 1.2, 2.6, 5.8, 12.8, 28.2, 62, 136.4, and 300 nM for 120 s, followed by dissociation for 900 s. Dissociation constants were determined based on fits applying a 1:1 interaction model. **b** Thermal protein stability measurements were performed using the NanoDSF technique. Proteins were diluted to 5 μM concentration in kinase buffer and incubated with 1, 10, or 100 μM abemaciclib for 10 min. Measurements were performed in duplicates ($n = 2$ biologically independent samples) and are depicted as mean. **c** Thermal protein stability measurements were performed as in (**b**) using ADP·Mg$^{2+}$, ATP·Mg$^{2+}$, or abemaciclib. ΔT$_m$ (max.–min. in °C) is depicted as mean. For raw data see Supplementary Table 3. Source data are provided as a Source Data file.

pocket in DYRK1A. These two structures of abemaciclib in complex with novel target proteins, exhibiting both similar efficacies and binding affinities, add to the understanding of the mode of action of this clinically used drug.

## Discussion

In this study, we identify the CMGC kinases DYRK1A and HIPK2/3 as potent target proteins of abemaciclib, exhibiting binding affinities and IC$_{50}$ values in the same range as those for CDK4 and CDK6. Off-target effects are a commonly observed phenomenon in drug pharmacology. As ATP-competitive inhibitors target a structurally and functionally similar binding site, poly-pharmacology is often seen and this target promiscuity can lead to both advantageous and detrimental therapeutic consequences[45]. Abemaciclib, reported to be a selective CDK4/6 inhibitor[40], is a third generation CDK-directed drug with an impressive clinical performance in the treatment of HR+ metastatic breast cancer[41–43]. However, it has been suggested early on that abemaciclib may exhibit pan-selective kinase inhibition properties that exceeds those of related CDK4/6 inhibitory drugs as palbociclib and ribociclib[43,46]. Transcriptional profiling on a panel of seven breast cancer cell lines showed indeed a characteristic signature of significantly downregulated genes for abemaciclib that goes beyond the profile of the target protein CDK4/6 kinase activity[46].

On a molecular level, the specificity of abemaciclib for CDK4/6 over CDK1/2/3/5 has been attributed in part to a residue at the entry site of the nucleotide binding pocket[44]. This residue (T107 in Cdk6) resides on the first turn of helix αD following the hinge region between the N- and C-lobe and is located nine amino acids downstream of the bulky gatekeeper residue in HIPKs and DYRKs. Its side chain is opposing the positively charged piperazine moiety of abemaciclib. While this 'entry guard' is a small, polar threonine in CDK4/6, it is a large positively charged lysine in CDK1/2/3/5, possibly leading to repulsion of the compound. Since other compound-interacting residues in the kinase active site are broadly similar, we suggest that this entry guard adopts a key role in the efficacy of target engagement by abemaciclib. Of note, the transcription elongation regulating kinases Cdk9, Cdk12, and Cdk13 contain a small glycine at this position, allowing the C-terminal extension helix to fold back onto the ATP-binding site[29].

Unexpectedly, the structure of HIPK3 adopts an active conformation with a fully accessible ATP-binding site even though the kinase is in its apo-state. This open conformation results from a stably ordered activation loop with the phosphorylated tyrosine integrated in an extensive salt-bridge and hydrogen-bond network with the αC helix acquiring the "in" position (Fig. 1d). A striking difference of the HIPK/DYRK family compared to CDKs relies in the phosphorylation of a tyrosine in HIPKs/DYRKs opposed to a threonine residue in CDKs for kinase activation.

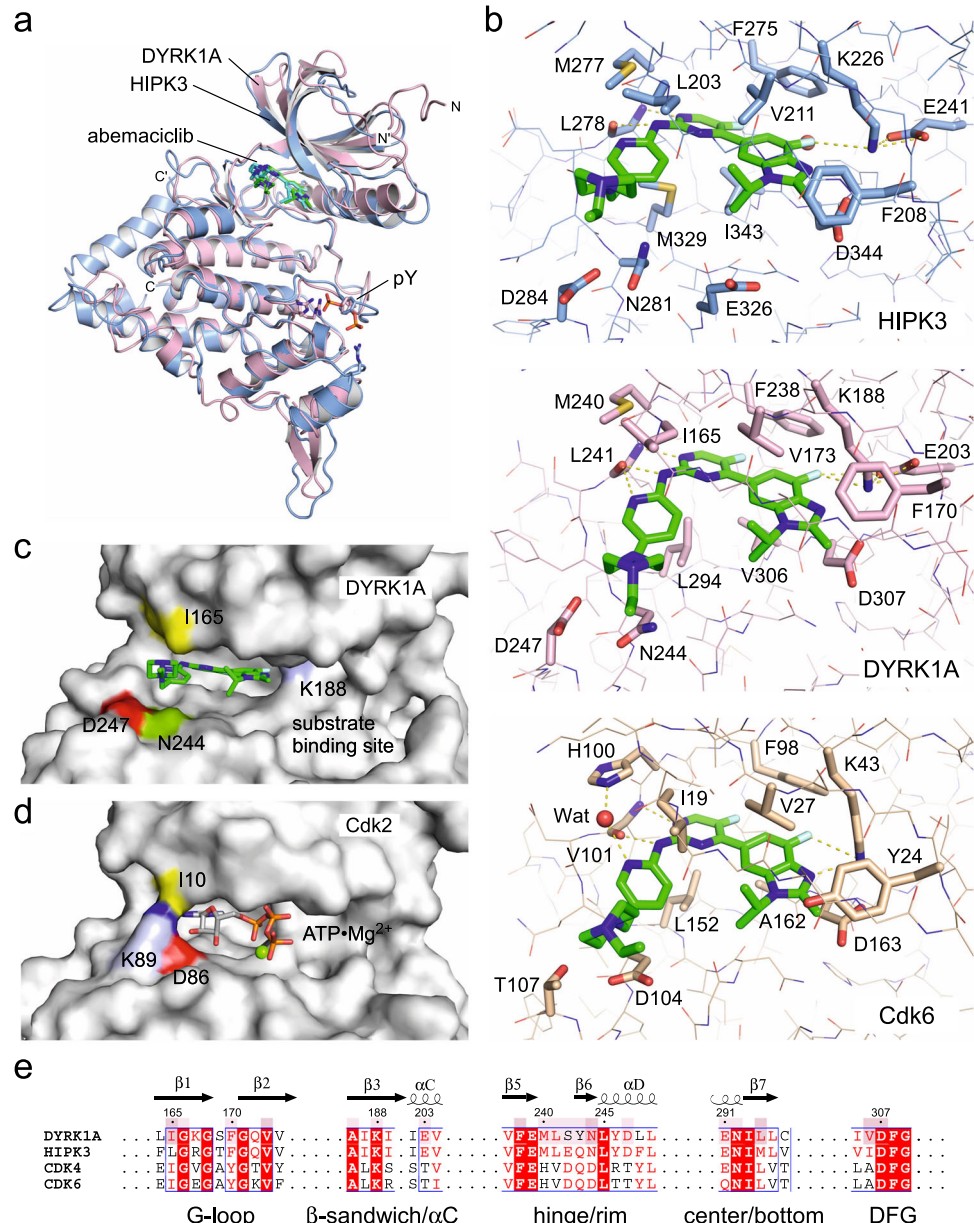

**Fig. 8 Complex structures of DYRK1A and HIPK3 with abemaciclib. a** Overlay of the kinase domains of DYRK1A–abemaciclib (rose/green; PDB code 7O7K) and HIPK3–abemaciclib (light blue/cyan; PDB code 7O7J). **b** Binding modes of abemaciclib to HIPK3, DYRK1A, and Cdk6. Overlay of HIPK3–abemaciclib (PDB 7O7J), DYRK1A–abemaciclib (PDB 7O7K) and Cdk6–abemaciclib (PDB 5L2S)[43]. Key active site residues are labeled, and hydrogen bonds are rendered as dotted lines. **c** Surface display of DYRK1A with space providing key residues N244 and D247 indicated. Isoleucine 165 is contributing with the largest buried surface area of all residues to the interaction with abemaciclib. **d** In CDK2 (PDB code 1QMZ)[19], K89 occupies the site where otherwise the piperazine ring of abemaciclib interacts with I165 of DYRK1A. This sequence variability might contribute to the interaction specificity of abemaciclib to CMGC kinases. **e** Sequence alignment of human DYRK1A, HIPK3, Cdk4, and Cdk6 in the interacting regions with abemaciclib. Residues in DYRK1A mediating direct interactions with abemaciclib are boxed light and dark rose according to the buried surface area (8–20 and >20 Å², respectively).

This change goes along with a lack of conservation of the HRD arginine, which is instead an alanine in HIPKs (HAD) and a cysteine in DYRKs (HCD). In most CDKs, this arginine forms a salt bridge with the T-loop threonine upon its phosphorylation. Together with an arginine of the αC helix (mostly REI) and the arginine of the DFGLAR motif, the coordination of three distant loops by salt bridges with the pT-site induces the conformational changes required for CDK activation[47]. HIPKs and DYRKs instead contain a R(F/Y)YR motif four residues downstream of the activating tyrosine, whose two arginines were shown to form salt bridges with the phosphate group of the auto-phosphorylated tyrosine. This holds for all structures of the DYRK family known

to date as well as for the one of HIPK2. For HIPK3 in contrast, we find for both structures determined in this study that the phosphate moiety of pY359 forms a salt bridge with R431 of the CMGC-specific insert region. While the conformation is stabilized by crystallographic contacts with a neighboring molecule, it shows nonetheless the conformational variability of the auto-activation site for stabilization of the active kinase.

Not only with respect to structural properties but also regarding function, HIPKs show some similarities with DYRK family members[6,8,20,48]. We found that HIPKs may be involved in the direct regulation of the transcription machinery as they phosphorylate the general transcription factors SPT5 and c-Myc,

as well as Ser2 and Ser5 residues of RNA pol II CTD in in vitro kinase assays. These findings are reminiscent to the description of DYRK1A as a transcriptional regulator by acting as a CTD kinase at gene loci, which are functionally associated with translation, RNA processing, and cell cycle control[26]. One of the first discovered hallmarks of HIPK1–3 was their localization to nuclear speckles by a speckle-retention signal of low complexity at the C-terminus[11,12,18]. Intriguingly, DYRK1A and P-TEFb both possess a low-complexity histidine-rich domain, which promotes binding and hyperphosphorylation of RNA pol II CTD in nuclear condensates, mediated by liquid–liquid phase separation[49]. It would therefore be interesting to examine in future studies, whether the localization of HIPKs to nuclear speckles is mediated by liquid–liquid phase separation and whether this is crucial for RNA pol II CTD hyperphosphorylation in vivo. Moreover, it has recently been shown that DYRK1A activity towards RNA pol II is modulated by the WD40-repeat protein DCAF7 at myogenic gene loci[27]. Interestingly, DCAF7 is a known interactor of HIPK2 regulating its kinase function in stress response[50,51], leading to the question whether this effect may also be a consequence of direct phosphorylation of RNA pol II and co-factors. Further studies are necessary to investigate whether HIPKs interact with the transcription machinery in vivo, and whether this is restricted to certain genes as it seems to be the case for DYRK1A.

Transcription regulating kinases have evolved as important drug targets for cancer therapies[52]. Even though HIPKs were identified as bona fide tumor suppressors in many cancer types, some studies point to the oncogenic potential of HIPKs[32,53,54]. Furthermore, members of this kinase family are involved in the pathology of Huntington´s disease, Alzheimer's disease, diabetes type II, and chronic fibrosis, raising the development of HIPK specific inhibitors to potentially valuable therapeutics[38,48]. With the crystal structures of apo HIPK3 as well as of HIPK3 and DYRK1A bound to abemaciclib determined, we provide the structural basis for a rational inhibitor design of this human kinase family. Whether abemaciclib's therapeutic success benefits from the simultaneous inhibition of the cell cycle regulating kinases Cdk4/Cdk6 in combination with off-target effects from the transcription kinases Cdk9, DYRK1A, and HIPKs awaits a detailed examination in future studies.

## Methods

**Protein expression**. Recombinant proteins were expressed in *Escherichia coli* BL21 (DE3) pLysS bacterial cells (DYRK1A, HIPK2, HIPK3, and GST-CTD[52], His-c-Myc, GST-SPT5, GST-RB1, GST-CycD3, and CycT1) or in baculovirus infected *Spodoptera frugiperda* 9 (*Sf9*) insect cells (HIPK1, HIPK2, HIPK3, MBP-HIPK4, Cdk4/CycD3, Cdk9, and Cdk12/CycK). For large scale expression in *E. coli*, a pre-culture supplemented with antibiotics was grown overnight at 37 °C that was subsequently used to inoculate LB media to an $OD_{600}$ of 0.1. Cells were grown at 37 °C to an $OD_{600}$ of 0.8, followed by induction of protein expression using 0.4 mM IPTG (isopropyl β-D-thiogalactoside) at 20 °C overnight. Cells were harvested by centrifugation, washed with PBS, and stored as pellets at −20 °C. Expression in *Sf9* insect cells was performed using the MultiBac^Turbo system[55]. Liquid cultures of *Sf9* cells were maintained at 27 °C in SF-900 III SFM medium (Invitrogen) at 80 rpm. Plasmids were transformed into *E. coli* (DH10) MultiBac^Turbo cells for T7-dependent integration into the baculoviral genome. Recombinant bacmid DNA was isolated, used for the initial transfection of *Sf9* cells, followed by further amplification of baculoviruses. For protein expression, *Sf9* cultures were infected by addition of 2% (v/v) of virus stock V2 at a density of $1.5 × 10^6$ cells per mL. Cells were harvested by centrifugation 72 h after infection, washed with PBS, and stored as pellets at −20 °C.

**Protein kinases**. The coding sequence of human wild-type HIPK1 (residues 154-554, UniProt accession number Q86Z02) and of human wild-type HIPK2 (residues 160–563, UniProt accession number Q9H2X6) were purchased as synthetic genes (GeneArt, Regensburg). HIPK1 was cloned into a modified pACEBac1 vector for expression in *Sf9* insect cells, containing an N-terminal MBP-affinity tag followed by a TEV (tobacco etch virus) protease cleavage site. HIPK2 was cloned into a pET-28a vector and a pACEBac1 vector both with an N-terminal GST-affinity tag followed by a TEV protease cleavage site. The coding sequence of the human

wild-type HIPK3 kinase domain (residues 159–562, UniProt accession number Q9H422) was PCR-amplified from Addgene plasmid #23467 and inserted into a pGEX-4T1 vector and a pACEBac1 vector, both containing an N-terminal GST-affinity tag followed by a TEV protease cleavage site. The human wild-type HIPK4 full-length coding sequence (residues 2–616, UniProt accession number Q3V016) was cloned from Addgene plasmid #23760 by PCR and ligated into a pACEBac1 vector containing an N-terminal MBP-tag with TEV protease cleavage site. For cloning of all four HIP kinases, *Eco*RI and *Not*I restriction sites were used. Kinase-dead mutants of HIPK1 (D315N), HIPK2 (D324N), HIPK3 (D322N), and HIPK4 (D136N) were generated by site directed mutagenesis and purified as described for wild-type protein. All expression plasmids used in this study were validated by DNA sequencing. If not stated otherwise, recombinant HIPKs used for experiments in this study were derived from *Sf9* insect cells, while HIPKs produced from *E. coli* were only used for autophosphorylation experiments.

For large scale purifications of MBP-HIPK1, GST-HIPK2, GST-HIPK3 and MBP-HIPK4, cells were resuspended in lysis buffer (50 mM Hepes pH 7.4, 500 mM NaCl, 1 mM β-mercaptoethanol (β-ME), 5% glycerol) and disrupted by sonication. The lysate was cleared by centrifugation in a Beckman-Coulter Avanti JXN-26 centrifuge at $50,000 × g$ for 45 min at 10 °C and applied to GSTrap FF columns or MBPTrap HP columns (Cytiva), equilibrated with lysis buffer using an ÄKTA prime chromatography system (GE Healthcare). Following extensive washes with 10 column volumes (CV) of lysis buffer, the protein was eluted in GST elution buffer (50 mM Hepes pH 7.4, 300 mM NaCl, 5% glycerol, 5 mM β-ME and 10 mM glutathione) or MBP elution buffer (50 mM Hepes pH 7.4, 300 mM NaCl, 5% glycerol, 5 mM β-ME and 10 mM maltose). Protein solutions were concentrated using Amicon Ultra centrifugal filters (Millipore) and digested with TEV protease 1:100 overnight at 4 °C for affinity tag removal. The MBP-HIPK4 protein construct was found to be only partially cleaved by TEV protease and therefore not digested. For SEC experiments, protein samples were loaded onto a preparative HiLoad 16/60 Superdex 200 prep grade column (Cytiva) equilibrated with SEC buffer (20 mM Hepes pH 7.4, 150 mM NaCl, 1 mM TCEP, 5% glycerol). Fractions of the main peak containing pure protein were analyzed by SDS-PAGE, pooled, and concentrated. Proteins were aliquoted, snap frozen in liquid nitrogen, and stored at −80 °C. For all kinases freeze-thaw cycles were avoided.

An expression plasmid of the human DYRK1A kinase domain (residues 127–485, UniProt accession number Q13627) was purchased from Addgene (#38913) in a pNIC28-Bsa4 backbone containing an N-terminal His$_6$ affinity tag followed by a TEV protease cleavage site. DYRK1A purification was performed as it is described here for HIPKs, only lysis buffer (50 mM Hepes pH 7.4, 300 mM NaCl, 5 mM β-ME, 5% glycerol, 25 mM imidazole) and elution buffer (50 mM Hepes pH 7.4, 500 mM NaCl, 5% glycerol, 5 mM β-ME, and 250 mM imidazole) differed. For affinity chromatography, HisTrap FF columns (Cytiva) were used and SEC was carried out using a preparative HiLoad 16/60 Superdex 75 prep grade column (Cytiva).

A plasmid encoding full-length human wild-type Cdk4 (residues 1–303, UniProt accession number P11802) was purchased from Origene (NM_000075), PCR-amplified with *Bam*HI/*Eco*RI restriction sites, and cloned into a pACEBac1 expression vector modified with an N-terminal His$_6$ affinity tag followed by a TEV protease cleavage site. Full-length human wild-type Cyclin D3 (residues 1–292, UniProt accession number P30281) was amplified from a HEK cDNA library and cloned into a pIDK vector, modified with an N-terminal GST-affinity tag followed by a TEV cleavage site using *Nco*I/*Kpn*I restriction sites. Both vectors were fused by in vitro Cre recombination and used for transformation of DH10 MultiBac^Turbo cells. The Cdk4/CycD3 complex was purified as described above via GST-affinity chromatography, followed by TEV protease cleavage and SEC. Human wild-type Cdk6/CycD3 protein complex was purchased from ProQinase (Freiburg).

Full-length, human wild-type Cdk9 (residues 1–372, UniProt accession number P50750) and human wild-type Cyclin T1 (residues 1–272, UniProt accession number O60563) cDNA constructs were a kind gift from Matija Peterlin (University of California at San Francisco, CA). CycT1 was PCR-amplified and inserted into a pGEX-4T1 expression vector containing a N-terminal GST-affinity tag followed by a TEV protease cleavage site using *Bam*HI/*Eco*RI restriction sites. CycT1 was expressed and purified from *E. coli* as described above for GST-affinity chromatography. Cdk9 was PCR-amplified and inserted into a pACEBac1 vector containing an N-terminal hexa-histidine affinity tag followed by a TEV protease cleavage site using *Eco*RI/*Not*I restriction sites. Cdk9 was expressed in *Sf9* insect cells and purified via His affinity chromatography as it was described for DYRK1A. After affinity chromatography, both proteins were concentrated, mixed in a 1:1 ratio and applied to SEC.

Human wild-type Cdk12 (residues 714–1063, UniProt accession number Q9NYV4) and human wild-type Cyclin K (residues 1–267, UniProt accession number O75909) were codon optimized for expression and synthesized by GeneArt (Regensburg). Cdk12 was cloned into modified pACEBac1 acceptor vector containing an N-terminal GST-affinity tag followed by a TEV protease cleavage site. CycK was cloned into a pIDK donor vector similarly modified by an N-terminal GST-TEV tag. Full-length CDK-activating kinase CAK1 (UniProt accession number P43568) from *S. cerevisiae* was cloned into a pIDC donor vector without any affinity tag. All three vectors were fused by in vitro Cre recombination, co-expressed in *Sf9* insect cells and purified by GST-affinity and SEC as described above.

**Substrate proteins**. Human wild-type CTD containing all 52 hepta-repeats of the RNA pol II subunit Rpb1 (residues 1587-1970, UniProt accession number P24928) was purchased from BioScience, UK (clone RPCIB753H14141Q) and cloned into a modified pGEX-6P1 expression vector using NcoI/EcoRI restriction sites, containing an N-terminal GST-affinity tag followed by a PreScission 3C protease cleavage site. Protein purification was carried out as described above, except that the GST-tag was not cleaved off but kept intact as GST-CTD[52] protein.

The coding sequence of human wild-type retinoblastoma-associated protein 1 (Rb1, UniProt accession number P06400) was purchased from Addgene (#82275). Its C-terminal domain (residues 761-928) was cloned into a pGEX-6P1 expression vector containing an N-terminal GST-affinity tag followed by a TEV protease cleavage site using BamHI/EcoRI restriction sites. Protein purification was performed as described above. For Rb1, the GST-affinity tag was not exposed to protease cleavage, instead recombinant protein was kept as intact GST-Rb1.

Human wild-type SPT5 protein (residues 748-1087, UniProt accession number O00267) was cloned from a cDNA library (GenBank accession code: DQ896795) into a pET-28a expression vector modified with an N-terminal GST-affinity tag followed by a TEV cleavage site using EcoRI/NotI restriction sites. Purification of SPT5 was carried out as described above for GST-fusion proteins using a preparative HiLoad 16/60 Superdex 75 prep grade column (Cytiva) for gel filtration. Protease cleavage was not performed, instead recombinant protein was kept as intact GST-SPT5.

Human wild-type c-Myc (residues 17-167, UniProt accession number P01106) was purchased as a cDNA clone (GenBank accession number BC000917.2) from imaGenes (BioScience). NcoI/EcoRI restriction sites were used for cloning c-Myc into a pProEx-HTa vector containing an N-terminal His₆ affinity tag followed by a TEV cleavage site. Recombinant His₆-c-Myc was purified from inclusion bodies. Protein expression was carried out in E. coli as described above overnight at 37 °C. After cell harvest, pellets were resuspended in resuspension buffer (50 mM Hepes pH 7.6, 100 mM NaCl, 5 mM β-ME), applied to sonication, and centrifuged at 30,000 × g and 4 °C for 30 min. Pellets were resuspended in 40 ml wash buffer (50 mM Hepes pH 7.6, 100 mM NaCl, 1 M urea, 0.5% Triton X-100, 5 mM β-ME) followed by centrifugation at 30,000 × g and 4 °C for 30 min. This step was performed three times. Afterwards, pellets were resuspended in 40 ml lysis buffer (50 mM Hepes pH 7.6, 100 mM NaCl, 5 mM β-ME). After another centrifugation step at 30,000 × g for 30 min at 4 °C, pellets were resuspended at room temperature in 40 ml extraction buffer (50 mM Hepes pH 7.6, 100 mM NaCl, 8 M Urea, 5 mM β-ME) for 60 min, followed by centrifugation at 30,000 × g for 30 min at 20 °C. Supernatant was dialyzed with refolding buffer A (50 mM Hepes pH 7.6, 100 mM NaCl, 4 M Urea, 5 mM β-ME) overnight at room temperature, followed by dialysis in refolding buffer B (50 mM Hepes pH 7.6, 100 mM NaCl, 5 mM β-ME, 10 mM imidazole) for 6 h at 4 °C. After centrifugation at 30,000 × g for 30 min at 4 °C, the protein was applied to HisTrap FF columns (Cytiva) equilibrated with refolding buffer B, followed by extensive washes with 5 CVs of refolding buffer B. Protein was eluted in elution buffer (50 mM Hepes pH 7.6, 100 mM NaCl, 5 mM β-ME, 350 mM imidazole). After protein concentration, SEC was performed as described above using a preparative HiLoad 16/60 Superdex 75 prep grade column (Cytiva).

**Crystallization and diffraction data collection**. Initial crystallization screens of HIPK3 were conducted using a homemade PegMix screen (0.1 M Hepes pH 7.0, 30% of medium weight PEG-mixture). Purified HIPK3 protein was concentrated to 10.5 mg/ml, mixed with 1 mM ADP/Mg²⁺ and crystallized by the hanging drop vapor diffusion method at 15 °C. Optimized hexagonal-shaped crystals (70 × 90 × 100 μm) were grown at a 1:1 ratio of protein and reservoir solution containing 0.1 M Hepes (pH 7.5), 0.2 M MgCl₂, and 15–17% (v/w) medium weight PEG mix.

Crystals of HIPK3 in complex with abemaciclib were obtained by mixing HIPK3 with a fivefold excess of abemaciclib, followed by incubation on ice for 30 min prior to crystallization. Each hanging drop was set using 1:1 ratio of protein-ligand mixture and mother-liquor. The initial crystals were optimized by micro-seeding. Optimized crystals at a size of 100 × 100 × 70 μm appeared in about 4 to 5 days at 15 °C in drops containing 0.1 M Tris/HCl (pH 8.0), 0.2 M MgCl₂, and 8% PEG 8000 solution.

DYRK1A protein in complex with abemaciclib was crystallized with 14–16% PEG 4000, 0.1 M sodium citrate (pH 6.5) and 0.1 M ammonium sulfate at a fivefold molar excess of the inhibitor. Crystals optimized for diffraction data collection (200 × 300 × 80 μm) appeared in about a week at 15 °C using the handing drop vapor diffusion method.

Crystals of apo-HIPK3 and the two complexes with abemaciclib were cryoprotected with 15–20% ethylene glycol in mother-liquor and flash frozen in liquid nitrogen. Diffraction data for all three samples were collected from a single loop-mounted crystal each, held in stream of liquid nitrogen gas at 100 K. The diffraction data sets (Supplementary Table 1) were collected at the PX1 synchrotron beamline at Swiss Light Source, Villigen, Switzerland, equipped with an Eiger detector.

**Structure determination and model building**. Data were processed and scaled using the XDS program package[56]. To correct for anisotropic scattering, HIPK3 apo diffraction data were submitted to the StarAniso server (Global Phasing Ltd). Ellipsoidal truncation resulting in low completeness did not hamper the available data resolution for model building and refinement[57,58]. The phase problem was solved by molecular replacement using PHASER[59]. The coordinates of HIPK2 (PDB 6P5S)[20] were used as search models for the apo structure of HIPK3. The model was refined by alternating cycles using PHENIX[60]. Manual rebuilding and visual comparisons were made using the graphical program COOT[61]. The stereo chemical quality of the model was confirmed using a Ramachandran plot. Protein interfaces and accessible surface areas were calculated with the program PDBe-PISA. Molecular diagrams were drawn using the PyMOL molecular graphics suite (Schrödinger, LLC). The final apo-HIPK3 structure includes one continuous chain of residues 184–550 but showed no ADP in the nucleotide binding site, while the structure of HIPK3 in complex with abemaciclib contains residues 184–551 and has been refined to $R_{work}$ and $R_{free}$ values of 24.3/27.3% and 24.6/27.5%, respectively. The structure of the DYRK1A–abemaciclib complex was solved by molecular replacement using the coordinates of DYRK1A (PDB 2VX3)[4]. A clearly defined region of electron density from a non-protein entity appeared at the ATP exit site that was modeled as two citrate molecules, exhibiting a central density in between both. Following a previous report of a crystal structure determined at 1.35 Å resolution[62], refinement of a Li⁺-bis-citrate complex gave a good fit to this electron density. The final model of DYRK1A in complex with abemaciclib contains residues 134–480 for both chains and has been refined to $R_{work}$ and $R_{free}$ values of 18.6/21.1%. Details of the diffraction data collection, structure quality, and refinement statistics are given in Supplementary Table 1.

**Small molecular compounds**. Small molecular compounds flavopiridol, (R)-roscovitine, and staurosporine as well as the reported compounds for kinase inhibition SCH727965 (dinaciclib), PD0332991 (palbociclib), LY2835219 (abemaciclib), GSK1059615, AS601245, and NVP-2 were purchased from MedChemExpress. Compounds XMD8-70, XMD8-62-i, JWD-065, HTH-01-091, CVM-05-145-3, and CVM-06-033-2 were developed and provided by the Gray laboratory.

**In vitro kinase assays**. For radioactive kinase assays, 0.2 μM kinase was pre-incubated with indicated concentrations of inhibitor and 200 μM ATP containing 0.45 mCi [³²P]-γ-ATP/mL (Perkin Elmer) in kinase buffer (150 mM HEPES (pH 7.6), 34 mM KCl, 7 mM MgCl₂, 2.5 mM dithiothreitol, 5 mM β-glycerol phosphate) for 5 min prior to starting the kinase reaction by addition of substrate at indicated concentrations. Reactions were incubated for 15 min at 30 °C and 300 rpm, and stopped by addition of EDTA to a final concentration of 50 mM. Samples were spotted onto Amersham Protran nitrocellulose membrane (GE Healthcare), followed by three washing steps for 5 min each with 0.75% (v/v) phosphoric acid. Counts per minutes were determined in a Beckman Liquid Scintillation Counter (Beckman-Coulter) for 1 min. Measurements were performed in duplicates and represented as mean with standard deviation (SD). GraphPad Prism (v.7) was used for data analysis and representation.

**Western blots**. For western blot analyses, kinase assays were performed with 0.2 μM kinase, 50 mM GST-CTD[52], and 200 μM ATP in kinase buffer (see above) at 300 rpm for 15 min. Reactions were stopped by addition of SDS sample buffer and subjected to 12% SDS-PAGE analysis. After transfer to Amersham Protran nitrocellulose membrane (GE Healthcare), monoclonal antibodies specific for RNA pol II CTD phosphorylation sites pTyr1 (3D12), pSer2 (3E10), pThr4 (G07), pSer5 (3E8), and pSer7 (4E12) were applied as described previously in a 1:100 ratio in 0.1% Tween in PBS[30]. A chicken anti-rat immunoglobulin G (IgG) horseradish peroxidase (HRP)-coupled antibody was applied as secondary antibody 1:5000 in 0.1% Tween in PBS. Membranes were incubated with ECL-solution for 1 min and then analyzed with a CCD camera ChemiDoc XRS+ system (BioRad).

**Protein thermal stability analyses**. Nano-differential scanning fluorimetry measurements were performed for the determination of thermal protein stability using a Prometheus NT.48 (NanoTemper) device operating at the two wavelengths $\lambda = 330$ and 350 nm. Proteins were diluted to 5 μM in kinase buffer (see above) and incubated with 1, 10, or 100 μM ADP, ATP, or abemaciclib for 10 min before measurement. The thermal stability was monitored from 20 to 70 °C at a heating rate of 1.5 °C/min using the PR.ThermControl software. Measurements were performed in duplicates.

**Surface plasmon resonance measurements**. SPR experiments were performed using a Biacore 8 K instrument (cytiva). All steps were performed at 25 °C. The system was equilibrated with running buffer (20 mM Tris pH 7.4, 150 mM NaCl, 0.1 mM MgCl₂, 0.05% Tween20, 2% DMSO). Kinases were immobilized in running buffer without DMSO using amine coupling. Before protein immobilization, flow cell 1 and 2 surfaces of a CM5 sensor chip were both activated for 15 s with 50 mM NaOH (30 μL/min), followed by activation with a 1:1 mixture of 0.1 M NHS (N-hydroxysuccinimide) and 0.1 M EDC (3-(N,N-dimethylamino) propyl-N-ethyl-carbodiimide) (10 μL/min) for 7 min. The flow system was washed with 1 M ethanolamine pH 8.0. Kinases were diluted 1:5 in acetate buffer (pH 5.5 for DYRK1A, Cdk4/CycD3, and pH 5.0 for HIPK2 and HIPK3). To obtain high levels of active immobilized kinase on the sensor chip surface, kinases were preincubated for 5 min at room temperature with a two-fold excess of abemaciclib prior to dilution in acetate buffer. Kinase immobilization was carried out on the flow cell

1 surface for 160 s at a flow rate of 10 μL/min. Subsequently, surfaces were blocked with 1 M ethanolamine pH 8.0 (10 μL/min) for 7 min. An abemaciclib wash out step was performed for at least 60 min (20 μL/min) before conducting kinetic measurements.

Kinetic binding of abemaciclib was measured as single-cycle kinetics. The compound was injected (30 μL/min, association: 120 s, dissociation: 900 s) over both flow cells at increasing concentrations. of 0.55, 1.2, 2.6, 5.8, 12.8, 28.2, 62, 136.4, and 300 nM. Data were collected at a rate of 10 Hz. The data were double referenced by blank cycle and reference flow cell subtraction. To compensate any potential effects of the DMSO, the data was solvent corrected. Processed data were fitted using a 1:1 interaction model using the Biacore Insight Evaluation Software (Cytiva).

**Mass spectrometry analyses.** For peptide mass fingerprint analysis, proteins were separated by SDS-PAGE analysis and stained with Coomassie brilliant blue. Protein bands of interest were excised and analyzed by mass spectrometry at the proteomics facility of the Max Planck Institute for Biophysical Chemistry in Göttingen. Quantitative mass analyses were performed with native protein samples at the BSRC Mass Spectrometry Facility at the University of St. Andrews by liquid chromatography–electrospray ionization-mass spectrometry. Mass spectrometry data was analyzed using Scaffold4 (v. 4.8.7). See also the Source Data.

**Reporting summary.** Further information on research design is available in the Nature Research Reporting Summary linked to this article.

## Data availability

The data supporting the findings of this study are available within the paper and its Supplementary Information Files, and are available from the corresponding author on reasonable request. Structure coordinates and diffraction data of the HIPK3 kinase domain in the apo form, the HIPK3–abemaciclib complex, and the DYRK1A–abemaciclib complex were deposited in the Protein Data Bank (http://www.pdb.org) under accession codes 7O7I, 7O7J, and 7O7K. Source data are provided with this paper.

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

## Acknowledgements

We would like to thank Jale Sariyar and Elif Tokmak for excellent technical assistance and Gregor Hagelueken for discussions. We also thank the beamline scientists at Swiss Light Source (SLS), Villigen, Switzerland, where diffraction data were recorded. We thank Prof. Kay Diederichs, University of Konstanz, for helpful discussion about HIPK3 native data and Prof. Henning Urlaub, Max Planck Institute for Biophysical Chemistry, Göttingen, and Sally Shirran, University of St. Andrews for mass spectrometry analysis. R.D. is funded by a grant from the Deutsche Krebshilfe (70114008). M.G. is funded by the Deutsche Forschungsgemeinschaft (DFG) under Germany's Excellence Strategy—EXC2151—390873048. This work was supported by a grant of the DFG to M.G. (GE 976/9-2).

## Author contributions

I.H.K. expressed and purified proteins and performed biochemical experiments. K.A. and I.H.K. crystallized proteins and determined the structures. J.M. supervised SPR experiments. J.W. and N.S.G. performed database searching and provided compounds. R.D., I.H.K., and M.G. conceptualized the study and I.H.K. and M.G. wrote the paper. All authors contributed to the final version of the paper.

## Funding

## Competing interests

N.S.G. is a founder, science advisory board member (SAB) and equity holder in Gatekeeper, Syros, Larkspur, Inception, C4, B2S, Allorion, Jengu, and Soltego (board member). The Gray lab receives or has received research funding from Novartis, Takeda, Astellas, Taiho, Janssen, Kinogen, Voronoi, Arbella, Deerfield, Epiphanes, and Sanofi. The other authors declare no competing interests.
