## [Peer Review File · Nature Communications]

Abemaciclib is a potent inhibitor of DYRK1A and HIP kinases involved in transcriptional regulationReviewers' Comments:

Reviewer #1:

Remarks to the Author:

The manuscript by Kaltheuner et al reports a thorough structure/function characterisation of related HIPK and DYRK kinases and shows that the Cdk4/6 inhibitor avidly inhibits HIPK and DYRKs. The work is explained well, executed to a high standard and represents a substantial body of research. However, my overriding impression is that the work is a bit disconnected, consisting of three separate components: a structural report of the HIPK3 structure, an auto- and substrate-phosphorylation characterisation of different HIPK and DYRK proteins, and then analysis of off-target inhibition of HIPKs by Abemaciclib. Each of these sections is informative, but the mechanistic link between them, and thus the clear overriding impact is somewhat unclear. I feel that some sort of biological validation is required to both show that the in vitro observations are functionally important and integrate these components into new mechanistic insight of HIPK3 (or HIPKs/DYRKs generally).

Major points

The functional relevance of the phosphorylation mapping, substrate profiling, and inhibition is somewhat unclear.

-Beyond the key tyrosine residue was there any mapping of phosphorylated residues on HIPKs onto available structures performed? Is there any functional relevance proposed here?

-The in vitro findings regarding substrate peptides is nice, but the title quite strongly implies a role in transcriptional regulation. To justify the title it would be good to see some validation of these targets in a biological context.

-Likewise the biological evidence linking abemaciclib, HIP kinases, and subsequently transcriptional elongation as implied in the title is relatively thin. Evidence of this type would tie the story into a coherent narrative.

-The authors compare differences between the known HIPK and DYRK structures and noted later in the discussion that there is a possible influence of crystal packing around this region. Are the observed differences proposed to mediate some functional difference? Is crystal packing influencing the related structures as well? Without showing that these differences have some effect it is not immediately clear what impact of the new HIPK3 structure is, relative to previous HIPK structures.

-While the authors show quite clean protein in Fig 3d, have any kinase dead mutants been tested in these assays? The caveat (particularly for HIPK1/4 that only show activity from insect cells) would be that would be that any trace kinase contamination could drive substrate phosphorylation.

-the inhibition and SPR, and structural data are quite compelling that abemaciclib targets HIPKs/DYRK, but the melting curves in Fig 6b are somewhat curious. While changes induced by abemaciclib are clear, is it normal for the ratio 350/330 ratio to decrease upon unfolding as for HIPK3? I have not used this particular instrument, but I would presume if tryptophan fluorescence is being tracked it would be more conventional for the ratio to increase. Could the protein be aggregating? Could be a minor point, but given most of the analysis in vitro uses recombinant protein this could be a relevant consideration.

Minor

Line 80, class1 instead of Class I elsewhere

Line 134/135, It is unclear what is meant by "yet, its N- and C-terminal extensions gained high protein solubility?"

Line 139, It is unclear how the two different models DYRK1A, and then HIP2K, were used for molecular replacement. Is this referring to HIP2K being used later for the ligand-bound structure? If so this is probably methods content rather than at this point of the results.

Figure 2 in the main text consisting of a full-length sequence alignment of the family members is probably not the best use of space. This would probably be best as a supplementary figure—I would propose that it is more important to show a structural comparison relative to other family members (current supplementary) in the main figures. Such a main figure could include excerpts of alignments in the structurally different regions.

Reviewer #2:

Remarks to the Author:

This manuscript presents the structures of apo homeodomain-interacting protein kinase (HIPK) 3 and of HIPK3 and DYRK1A bound to abemaciclib. Abemaciclib is currently in clinic for the treatment of a molecularly defined subset of breast cancer patients where it inhibits CDK4 and CDK6 activity. Biochemical and biophysical assays accompany the structural study to provide evidence that abemaciclib inhibits in vitro HIPK2, HIPK3, and DYRK1A activity towards three key proteins that regulate RNA transcription and in the case of DYRK1A with similar efficacy to its inhibition of CDK4 and CDK6. The paper includes a comprehensive comparative structural description of selected HIPK and DYRK family members, and of the selected inhibitor binding modes to different protein kinases. Taken together the authors provide a sound case for further studies to explore the hypothesis that abemaciclib might derive some of its patient efficacy from targeting DYRK1A in addition to CDK4 and CDK6. Given the widening clinical use of abemaciclib and other CDK4/6 inhibitors the findings will be of general interest to a wide audience.

Overall, the paper is tightly written, detailed and with an engaging narrative style. The structure descriptions are organised logically, clearly illustrated and informative of the key structural differences revealed by a comparison of HIPK3 and its close relatives. The biochemical and biophysical characterisations are carefully carried out with in most cases appropriate controls and number of repeats. The data provides a detailed structural understanding of abemaciclib binding to different protein kinase targets. Importantly it does not over-interpret the data from the careful structural, biochemical and biophysical studies by suggesting that in vivo efficacy follows.

I would recommend the manuscript is suitable to be considered for publication in Nature Communications subject to review of the suggestions made below.

Major Points:

1. Figure 4 and results at circa line 260. Comparison of HIPK1-4 and DYRK1A activity towards a panel of modified CTD peptides. This significant conclusion from this set of experiments is that “in contrast to P-TEFb, Cdk12 or Cdk13, a preference of pre-phosphorylated CTD substrates over consensus CTD was not observed” To strengthen this functional difference, P-TEFb, CDK12 or CDK13 should be included in the analysis as a control (and as a 6th panel in Figure 4e) to confirm that using this in vitro assay format functional differences reported in the literature are recapitulated.

Minor Points

1. Figure 1. Could all secondary structural elements mentioned in the text be included in panel c. Could the location of the ATP binding site be identified (for non-specialists) in panel d.
2. SuppFigure 2a. Label helix α N.
3. Figure 3d, include number of replicates and description of error bars (and review for other figures as appropriate).
4. Results, line 237. Statement “Interestingly, we found that the kinase activity of DYRK1A as well as

HIPK1-3 towards the C-terminal region of SPT5 is even more pronounced than of P-TEFb." Implicit as written, this statement suggests that the activities of DYRK1A and HIPK1-3 towards the selected substrates suggests more biological significance than the activity of CDK9-cyclin T. Given that the assay is an in vitro one and does not employ full-length constructs, perhaps this statement could be modified to "Interestingly, under the invitro conditions employed, we found that the kinase activity of DYRK1A as well as HIPK1-3 towards the C-terminal region of SPT5 is even more pronounced than of P-TEFb" (or similar).

5. Figure 4c,d. Please identify the positions of the hypo- (IIa) and hyper-phosphorylated (IIo) states of the GST-CTD substrate on the gels.

6. Results Line 250. Statement "Interestingly, it seems that the shift of the hyperphosphorylated GST-CTD[52] IIo band is provoked by Ser7 phosphorylation as seen for HIPK4 compared to the HIPK1-3 mediated Ser2 and Ser5 phosphorylation." Please clarify, the band identities are not mapped to specific phosphorylation events. The statement "provoked by S7 phosphorylation" suggests S7 is a priming event for subsequent phosphorylations at other sites. However, could it not just result from the more efficient phosphorylation of S7 by HIPK4 without having to evoke a particular mechanism? Though this point is addressed in the following paragraph, it might be helpful to review narrative here.

7. Methods, line 488. Please convert to xg or provide rotor ID.

Reviewer #3:

Remarks to the Author:

The manuscript "Abemaciclib is a potent inhibitor of DYRK1A and HIP kinases regulating transcriptional elongation" by Kaltheuner et al. describes a series of biophysical studies of the human HIPKs. The authors determined the first crystal structure in literature of HIPK3 kinase domain, and assessed phosphorylation state and the kinase activity of the recombinant protein samples. The works revealed that HIPKs phosphorylate the CTD of RNA pol II CTD in vitro. The author carried out a broad search against kinase inhibitors and identified Abemaciclib as a potent inhibitor of HIPK2, HIPK3, and DYRK1A. The result was further supported by the crystal structures of HIPK3 and DYRK1A bound to Abemaciclib. The data presented in this manuscript are mostly solid, with one caveat related to the heterogenous phosphorylation state of the protein samples (see comments below). This work is potentially interesting to broad audience, including structural biologists, medicinal chemists, and related biologists. Following are questions and comments for the authors.

- Table S2 shows heterogenous phosphorylation state of the recombinant protein samples. Since the Methods section shows no detailed steps of separating those species, this reviewer assumes that all subsequent studies were done with heterogeneously phosphorylated samples. First, phosphorylation occurred in expression hosts, especially in the insect cells, do not necessarily represent bona fide phosphorylation sites in vivo. Second, it is important to establish good understanding of the functional relevance of each site. Third, consider mapping the phospho-sites on to the crystal structure and evaluate whether phosphorylation influenced the substrate binding and recognition? and/or the inhibitor activity? Is there visible electron density in the crystal structures that correspond to any of the phospho-site?
- This reviewer suggests to reorganize the manuscript to better reflect the title. For example, moving apo-HIPK3 structure to a later part, so that it goes hand-in-hand with the inhibitor bound structures. In the discussion of crystal packing, please include its potential influence to compound binding interactions.
- The data processing statistics for apo-HIPK3 apo looks rather poor at 2.50Å resolution. Please consider more stringent resolution cutoff.

Detailed point-to-point reply to the Reviewers' comments:

Reviewer #1 (Remarks to the Author):

The manuscript by Kaltheuner et al reports a thorough structure/function characterisation of related HIPK and DYRK kinases and shows that the Cdk4/6 inhibitor avidly inhibits HIPK and DYRKs. The work is explained well, executed to a high standard and represents a substantial body of research. However, my overriding impression is that the work is a bit disconnected, consisting of three separate components: a structural report of the HIPK3 structure, an auto- and substrate-phosphorylation characterisation of different HIPK and DYRK proteins, and then analysis of off-target inhibition of HIPKs by Abemaciclib. Each of these sections is informative, but the mechanistic link between them, and thus the clear overriding impact is somewhat unclear. I feel that some sort of biological validation is required to both show that the in vitro observations are functionally important and integrate these components into new mechanistic insight of HIPK3 (or HIPKs/DYRKs generally).

Major points

The functional relevance of the phosphorylation mapping, substrate profiling, and inhibition is somewhat unclear.

-Beyond the key tyrosine residue was there any mapping of phosphorylated residues on HIPKs onto available structures performed? Is there any functional relevance proposed here?

We felt that the phosphorylation section was indeed the weakest part of the previous version of the manuscript. To make our intentions with that section and associated experiments easier to understand and correlate this to previous publications that described kinase regulating phosphorylations, we re-wrote the text and included now electrospray ionization-MS spectra of intact proteins both from *Sf9* insect cell and *E.coli* bacterial cell expression (new Figure 4b). From the intact masses, we identify indeed up to nine phosphorylations for HIPK2 expressed in *E.coli*, which are, however, not at stoichiometric levels. The more homogenous phosphorylation pattern results instead from *Sf9* cell expression with mostly one and a small proportion of a second phosphorylation for HIPK1 and HIPK3. The intact mass of HIPK4 could unfortunately not be determine as a protein construct with the kinase domain only did not work in our hands, and the MBP-HIPK4 full length protein used has a total mass of 111.9 kDa, which is beyond an accurate detection limit to determine the +80 Da for a phosphorylation.

By determining phospho-sites in the recombinant proteins using peptide mass fingerprint analyses (Suppl. Table 2), we aimed at identifying the critical residues that might be of importance for kinase activity and could explain the different activities we see in Figure 3 for recombinant proteins originating from *E.coli* (only auto-phosphorylated kinases) versus *Sf9* insect cells (auto-phosphorylated and phosphorylation from foreign kinases possible). To visualize these phospho-sites on the structure of HIPK3, we modelled an AlphaFold-X-ray fusion structure, as many phosphorylations were found in the N-terminus of the protein construct used (159-562) that was not resolved in the electron density map of the crystal structure (184-550). These mapping of all phosphorylation sites in our assays for HIPK1 to HIPK4 is now displayed in the new Supplementary Figure 3.

-The *in vitro* findings regarding substrate peptides is nice, but the title quite strongly implies a role in transcriptional regulation. To justify the title, it would be good to see some validation of these targets in a biological context.

-Likewise, the biological evidence linking abemaciclib, HIP kinases, and subsequently transcriptional elongation as implied in the title is relatively thin. Evidence of this type would tie the story into a coherent narrative.

We fully agree with the Reviewer that the substrate targets found for HIPK3 kinase phosphorylation in transcription should be validated next in a biological context. However, these experiments are hampered by the lack of a selective inhibitor targeting HIPK3 as HIP kinases have been shown to compensate each other in knock-out experiments with mice (Isono, et al., *Mol. Cell Biol.* 2006). As a follow-up study, we aim to look at the biological relevance of HIPKs generating analogue sensitive kinase mutants for substrate screenings, also in regard to their role in transcriptional regulation and cancer. However, the positive feedback of all reviewers regarding our *in vitro* experiments and the impact of the clinical inhibitor abemaciclib targeting HIPKs and DYRK1A encouraged us to submit our *in vitro* results in the current form as a self-contained study.

-The authors compare differences between the known HIPK and DYRK structures and noted later in the discussion that there is a possible influence of crystal packing around this region. Are the observed differences proposed to mediate some functional difference? Is crystal packing influencing the related structures as well? Without showing that these differences have some effect it is not immediately clear what impact of the new HIPK3 structure is, relative to previous HIPK structures.

We describe the HIPK3 apo structure and make the observation that there are differences around the region with the pY of the activation loop compared to the already existing crystal structure of HIPK2. A weak salt bridge of the pY to a symmetry related molecule is formed while an aspartate of this symmetry mate occupies the space of the pY instead.

Based on the Reviewers' comment, we double-checked for the other known HIP/DYRK structures, which are HIPK2 (6P5S), DYRK1A (2VX3), DYRK2 (3K2L) and DYRK3 (5Y86). Interestingly, in HIPK2 and DYRK3, but not DYRK1A or DYRK2, the pY of the activation loop is involved in crystal packing contacts. In HIPK2, the very N-terminus of helix α H is directly contacting pY with side chains of the sequence T₄₈₃DLE not being resolved in the crystallographic map (see picture below). This association is indeed similar to the one seen in our HIPK3 structure—albeit at a different conformation—as it is also the beginning of the α H helix that interacts with an adjacent molecule.

In DYRK3, it is a helical bundle preceding the N-lobe that directly contacts the phosphorylated tyrosine of the activation loop of a symmetry-related molecule (see picture below). However, for both structures holds that the pY is interacting with the RYYR residues and rather pushed into this conformation by the crystal contact than pulled out of this ionic interaction.

We now discuss these observations in light of the additional phosphorylation residues, mentioned by this and the third Reviewer, three residues downstream of the pY site (S362 in HIPK3) that was found in HIPK2 to be required for activity. This residue could readily interact with R₃₆₃YYR (which is in a tight 3₁₀ helix) and thus drive the pY side in the position observed in our crystal structure. Such a double phosphorylation pattern in the activation loop (pY₃₅₉)xx(pS)RYYR is reminiscent to the dual phosphorylations seen *e.g.*, in Cdk7, and might certainly impact the substrate specificity and overall kinase activity. Such analyses, however, would be beyond our current study but will certainly drive the HIP kinase field forward.

-While the authors show quite clean protein in Fig 3d, have any kinase dead mutants been tested in these assays? The caveat (particularly for HIPK1/4 that only show activity from insect cells) would be that any trace kinase contamination could drive substrate phosphorylation.

We fully understand the Reviewers' concerns that contaminations from the protein purification process could falsify the kinase activity measurements. We therefore cloned and purified kinase dead mutants for all four HIPKs from *Sf9* insect cells and *E.coli* proteins, always exchanging the catalytic D to N. Activity assays with these kinase dead mutants show no activity, excluding the possibility of any foreign kinase impurities as reason for the differences in activity. These new data are now incorporated in Figure 4a of the revised manuscript. A Coomassie-stained SDS PAGE analysis showing the purity and homogeneity of the kinase dead protein mutants is shown on the right.

-the inhibition and SPR, and structural data are quite compelling that abemaciclib targets HIPKs/DYRK, but the melting curves in Fig 6b are somewhat curious. While changes induced by abemaciclib are clear, is it normal for the ratio 350/330 ratio to decrease upon unfolding as for HIPK3? I have not used this particular instrument, but I would presume if tryptophan fluorescence is being tracked it would be more conventional for the ratio to increase. Could the protein be aggregating? Could be a minor point, but given most of the analysis in vitro uses recombinant protein this could be a relevant consideration.

The determination of thermal protein stability was performed using the nano-differential scanning fluorimetry with a NanoTemper Prometheus NT.48 device. The technology is based on the exposure or masking of internal tryptophane under native conditions by detecting changes in its intrinsic fluorescence during a thermal ramp. A change of the intrinsic tryptophane fluorescence can be both red or blue shifted depending on the changes of the microenvironment. The Trp fluorescence emission maximum is at around 330 nm when located in an apolar environment. In a polar environment, the Trp emission intensity usually decreases due to static and dynamic quenching by solvent molecules, and the peak emission is red-shifted to about 350 nm (Duy and Fitter, 2006). In addition, tryptophan fluorescence is strongly influenced by the proximity of other residues (*i.e.*, nearby protonated groups such as Asp or Glu that can cause quenching). Both protein constructs used in the experiment contain four Trp's of which two are conserved and may behave similar in both proteins (W383 and W444 following the HIPK3 sequence, Fig. 2), while at least one other Trp is on the surface of each protein. We have derived the original 350 nm and 330 nm fluorescence traces from the experiment.

When performing the nanoDSF experiment, both proteins were precipitated at 70°C which precluded a possible re-folding experiment going back from 70°C to 20°C. It is indeed very attentive to assume that HIPK3 denaturates into an aggregated state, while DYRK1A may stay longer upon thermal unfolding. Such a hypothesis would at least be consistent with our observation that HIPK3 loses its activity relatively quickly after 1 or 2 freeze-thaw cycles, while DYRK1A seems to be much more stable. We thank the reviewer for this attentive comment and will follow up this idea in future experiments with HIP kinases.

Duy, C., and Fitter, J. (2006). How aggregation and conformational scrambling of unfolded states govern fluorescence emission spectra. *Biophys. J.* **90**, 3704–3711.

Minor

Line 80, class1 instead of Class I elsewhere.

Thanks, corrected.

Line 134/135, It is unclear what is meant by "yet, its N- and C-terminal extensions gained high protein solubility?"

We agree with the reviewer that the statement of this sentence is unclearly worded. We now revised the sentence to: "This construct extends N- and C-terminally beyond the canonical kinase domain, but showed high solubility (Fig. 1a, b)." Since we report later on the (auto-) phosphorylation sites of HIP kinases, we would like to emphasize that the boundaries of the construct extend beyond the minimal kinase domain, especially since many phosphorylation marks are found in the N-terminal region preceding the kinase N-lobe (new Suppl. Fig. 3).

Line 139, It is unclear how the two different models DYRK1A, and then HIP2K, were used for molecular replacement. Is this referring to HIP2K being used later for the ligand-bound structure? If so, this is probably methods content rather than at this point of the results.

Yes, we agree that the description of the molecular replacement procedure to gain the crystallographic phases is Methods content rather than original data of the Results section. When we got the first diffraction data of the initial crystals of apo-HIPK3 (at resolutions around 3.5 Å) we successfully used the coordinates of DYRK1A (PDB accession code 2VX3) for molecular replacement as the closest homologue to HIPK3. However, when we recorded the 2.5 Å dataset of apo-HIPK3 the structure of HIPK2 (6P5S) just became available and we used this one for molecular replacement. We now only refer to the latter strategy and described this correctly in the "Structure determination and model building" paragraph of the Methods section.

Figure 2 in the main text consisting of a full-length sequence alignment of the family members is probably not the best use of space. This would probably be best as a supplementary figure—I would propose that it is more important to show a structural comparison relative to other family members (current supplementary) in the main figures. Such a main figure could include excerpts of alignments in the structurally different regions.

We agree with the Reviewer that the structural comparison relative to other family members as DYRK1A, DYRK2, DYRK3 and HIPK2 is very informative and contributes to an understanding of the divergence of these kinases. Since we included now new phosphorylation data by the masses of intact protein and mapped these sites to the kinases, we reorganized Figures 2 and 3 and included also the structural comparison as a new figure in the main text.

Reviewer #2 (Remarks to the Author):

This manuscript presents the structures of apo homeodomain-interacting protein kinase (HIPK) 3 and of HIPK3 and DYRK1A bound to abemaciclib. Abemaciclib is currently in clinic for the treatment of a molecularly defined subset of breast cancer patients where it inhibits CDK4 and CDK6 activity. Biochemical and biophysical assays accompany the structural study to provide evidence that abemaciclib inhibits in vitro HIPK2, HIPK3, and DYRK1A activity towards three key proteins that regulate RNA transcription and in the case of DYRK1A with similar efficacy to its inhibition of CDK4 and CDK6. The paper includes a comprehensive comparative structural description of selected HIPK and DYRK family members, and of the selected inhibitor binding modes to different protein kinases. Taken together the authors provide a sound case for further studies to explore the hypothesis that abemaciclib might derive some of its patient efficacy from targeting DYRK1A in addition to CDK4 and CDK6. Given the widening clinical use of abemaciclib and other CDK4/6 inhibitors the findings will be of general interest to a wide audience.

Overall, the paper is tightly written, detailed and with an engaging narrative style. The structure descriptions are organised logically, clearly illustrated and informative of the key structural differences revealed by a comparison of HIPK3 and its close relatives. The biochemical and biophysical characterisations are carefully carried out with in most cases appropriate controls and number of repeats. The data provides a detailed structural understanding of abemaciclib binding to different protein kinase targets. Importantly, it does not over-interpret the data from the careful structural, biochemical and biophysical studies by suggesting that in vivo efficacy follows.

I would recommend the manuscript is suitable to be considered for publication in Nature Communications subject to review of the suggestions made below.

Major Points:

1. Figure 4 and results at circa line 260. Comparison of HIPK1-4 and DYRK1A activity towards a panel of modified CTD peptides. This significant conclusion from this set of experiments is that “in contrast to P-TEFb, Cdk12 or Cdk13, a preference of pre-phosphorylated CTD substrates over consensus CTD was not observed” To strengthen this functional difference, P-TEFb, CDK12 or CDK13 should be included in the analysis as a control (and as a 6th panel in Figure 4e) to confirm that using this in vitro assay format functional differences reported in the literature are recapitulated.

We fully agree with the Reviewer on the completion of the HIP, DYRK and transcriptional CDK kinases for a comparison of RNA polymerase II CTD phosphorylations and their specificity for particular pre-phosphorylation marks. In the revised manuscript, we included measurements with P-TEFb and Cdk12/CycK as a quality control for kinase activities towards CTD pre-phosphorylated peptides (new Fig. 4e). In accordance with previous data, P-TEFb shows the highest activity towards consensus RNA pol II CTD, but also phosphorylates pS7 and K7 alterations, whereas Cdk12/CycK is highest active on a pre-phosphorylated Ser7 CTD. The comparison of these data now shows very impressively, that all HIP kinases 1–4 as well as DYRK1A do not develop any activity on pS7 CTD, whereas Cdk9 and Cdk12 do. Despite the (S/T)P consensus motif for the CMGC kinase group, this observation underlines the specificity of the different kinases.

Minor Points

1. Figure 1. Could all secondary structural elements mentioned in the text be included in panel c. Could the location of the ATP binding site be identified (for non-specialists) in panel d.

Yes, all secondary structure elements that are mentioned in the text are now included in panel c, as well as the location of the ATP binding site in panels c and d.

2. SuppFigure 2a. Label helix α N.

We thank the Reviewer for this attentive comment and now labelled helix α N in the Supplementary Figure 2a and indicated the two molecules of the crystallographic contact.

3. Figure 3d, include number of replicates and description of error bars (and review for other figures as appropriate).

We checked all figure legends again and included the number of replicates and description of error bars in the revised manuscript.

4. Results, line 237. Statement “Interestingly, we found that the kinase activity of DYRK1A as well as HIPK1-3 towards the C-terminal region of SPT5 is even more pronounced than of P-TEFb.” Implicit as written, this statement suggests that the activities of DYRK1A and HIPK1-3 towards the selected substrates suggests more biological significance than the activity of CDK9-cyclin T. Given that the assay is an *in vitro* one and does not employ full-length constructs, perhaps this statement could be modified to “Interestingly, under the *in vitro* conditions employed, we found that the kinase activity of DYRK1A as well as HIPK1-3 towards the C-terminal region of SPT5 is even more pronounced than of P-TEFb” (or similar).

We have rewritten the sentence accordingly to make the statement more precise. It is certainly true that distal elements could affect the kinase activity and its substrate specificity. The restriction to *in vitro* conditions thus clarifies the statement.

5. Figure 4c,d. Please identify the positions of the hypo- (IIa) and hyper-phosphorylated (IIo) states of the GST-CTD substrate on the gels.

We now labelled the hypo- and hyper-phosphorylated states of the GST-CTD substrate in Figure 4 with 'IIa' and 'IIo' as suggested by the Reviewer.

6. Results Line 250. Statement “Interestingly, it seems that the shift of the hyperphosphorylated GST-CTD[52] IIo band is provoked by Ser7 phosphorylation as seen for HIPK4 compared to the HIPK1-3 mediated Ser2 and Ser5 phosphorylation.” Please clarify, the band identities are not mapped to specific phosphorylation events. The statement “provoked by S7 phosphorylation” suggests S7 is a priming event for subsequent phosphorylations at other sites. However, could it not just result from the more efficient phosphorylation of S7 by HIPK4 without having to evoke a particular mechanism? Though this point is addressed in the following paragraph, it might be helpful to review narrative here.

Yes, this sentence is indeed misleading and does not express the direct observation of the different running behaviour of nascent pre-phosphorylated Ser7 CTD compared to nascent pre-phosphorylated Ser5 or Ser2 CTD before becoming a hyper-phosphorylated CTD in the time course experiment. This different migration behaviour can be only seen for HIPK4 at time points 15 min, 30 min and 45 min when comparing immuno-stained pSer2 and pSer5 bands with the pSer7 band. Whereas the anti-pSer7 antibody marks a higher band, the anti-pSer2 and -pSer5 antibodies are still at lower levels. This could result, for example, from a different kinase phosphorylation mechanism, *e.g.*, a more processive versus a more distributive phosphorylation pattern, or from a different structural conformation of the pSer7 versus the pSer5/pSer2 phosphorylated CTD. However, we did not intend to say that there could be a pSer7 priming effect which subsequently leads to pSer5/pSer2 phosphorylation by the same kinase. We therefore deleted the sentence. This whole observation is more for CTD aficionados, anyway.

7. Methods, line 488. Please convert to xg or provide rotor ID.

Yes, we fully agree with the reviewer and have changed all centrifugation data to *g*-force numbers in the revised version of the manuscript.

Reviewer #3 (Remarks to the Author):

The manuscript “Abemaciclib is a potent inhibitor of DYRK1A and HIP kinases regulating transcriptional elongation” by Kaltheuner et al. describes a series of biophysical studies of the human HIPKs. The authors determined the first crystal structure in literature of HIPK3 kinase domain, and assessed phosphorylation state and the kinase activity of the recombinant protein samples. The works revealed that HIPKs phosphorylate the CTD of RNA pol II CTD *in vitro*. The author carried out a broad search against kinase inhibitors and identified Abemaciclib as a potent inhibitor of HIPK2, HIPK3, and DYRK1A. The result was further supported by the crystal structures of HIPK3 and DYRK1A bound to Abemaciclib. The data presented in this manuscript are mostly solid, with one caveat related to the heterogeneous phosphorylation state of the protein samples (see comments below). This work is potentially interesting to broad audience, including structural biologists, medicinal chemists, and related biologists. Following are questions and comments for the authors.

- Table S2 shows heterogeneous phosphorylation state of the recombinant protein samples. Since the Methods section shows no detailed steps of separating those species, this reviewer assumes that all subsequent studies were done with heterogeneously phosphorylated samples. First, phosphorylation occurred in expression hosts, especially in the insect cells, do not necessarily represent bona fide phosphorylation sites *in vivo*. Second, it is important to establish good understanding of the functional relevance of each site. Third, consider mapping the phospho-sites on to the crystal structure and evaluate whether phosphorylation influenced the substrate binding and recognition? and/or the inhibitor activity? Is there visible electron density in the crystal structures that correspond to any of the phospho-site?

It is correct as the Reviewer suggests that we did not perform any further steps of separating the recombinant HIP kinases by their phosphorylation signatures, *e.g.*, through ion exchange chromatography (**first comment**). Following our previous experiences with transcriptional Cdk preparations, we first expressed HIPK3 in *Sf9* cells but also tried expression in *E.coli* (based on the reports of DYRK1A purifications) as a time- and cost-saving alternative. After figuring out the most suitable domain boundaries for stability and solubility, we quickly succeeded in growing protein crystals of HIPK3 from *Sf9* cells. This observation we considered itself as a proof of homogeneity and construct integrity. However, after this Reviewers' comments, we determined the atomic molecular masses of the intact proteins for HIPK1-3 from *E.coli* or *Sf9* cell expression by electrospray ionization-MS spectra to get an idea of the heterogeneity of the total phosphorylation marks. These new data are now presented in Figure 4b of the revised

manuscript. To our surprise, the *E.coli* bacterially expressed proteins showed a significantly higher number of total phosphorylations than the *Sf9* eukaryotic cell expressed proteins. We attribute this observation to an increased degree of auto-phosphorylation of the *E.coli* expressed protein and possibly phosphatase activity in *Sf9* cells. The highest number of phosphorylations was indeed found for HIPK2 from *E.coli* with up to 9 phospho-groups. The most homogeneous pattern was instead found for HIPK1 with mostly just one phosphorylation followed by HIPK3 with maximally two phosphorylations, both from insect cells. The peptide mass fingerprint spectrometry analysis summarized in Supplementary Table 2 is therefore only a hint to identify phosphorylated sites and the number of phosphorylated peptides detected versus the peptides flying just a rule of thumb for the absolute stoichiometry of phosphorylation.

We also agree with the Reviewer that the identified phosphorylation sites in HIPKs (depicted in Suppl. Table 2) resulting from expression in *Sf9* or *E.coli* cells do not necessarily correspond to *in vivo* phosphorylation sites in human tissue (**first point**). Instead, we were curious whether phosphorylation sites apart from the critical tyrosine residue were essential for kinase activity. For HIPK2, the *E.coli* protein with significantly more phosphorylations as the *Sf9* cell protein was slightly more active for c-Myc substrate phosphorylation. The characterization of the functional relevance of each individual phosphorylation site would be certainly very important (**second point**). However, such analyses are beyond the focus of this study and would require an enormous effort. There are already some high-quality studies published on the characterization of specific HIPK2 phosphorylations including a second phosphorylation site in the kinase activation loop. These studies are cited in the revised version of the manuscript.

We appreciate the Reviewers' comment on the mapping of the phospho-sites onto the structure of HIPK3 to visualize the localization and accessibility of these PTMs (**third point**). Following this suggestion, we generated a new figure, mapping the sites of all phosphorylations found in the peptide mass fingerprint analysis of HIPK1 to 4 on an AlphaFold–crystal structure fusion model to localize these sites (new Supplementary Figure 3). Modelling of the AlphaFold–X-ray fusion structure was necessary, as many phosphorylations were found in the N-terminus of the protein construct that was not resolved in the crystal structure.

Regarding the influence of substrate binding and recognition: Besides the critical tyrosine phosphorylation (Y359 in HIPK3), two additional phosphorylations were found in the

activation loop of HIP kinases. On the one hand, this is a serine two amino acids upstream of the pY motif (found in HIPK3), and on the other hand a serine three residues downstream of the pY motif (found in HIPK4). This latter phosphorylation site caught our attention as it has been described in HIPK2 to be required for full activity (Saul et al., 2013). Going back to the original MS data, we found that 4% of all flying peptides contained this phosphorylation also in our HIPK2 preparation (now also listed in Suppl. Table 2). This phosphorylation mark is directly preceding the HIP/DYRK-defining RYYR motif (pYxxpSRYYR) and could perfectly interact with the two arginines, thus displacing the pY in HIPK2 from this site. However, we think that concrete experiments regarding the phosphorylation preference for putative substrates is beyond the focus of this study, and the significance for this phosphorylation in HIPK2 has been already analysed in previous studies which are cited in the manuscript. Of note: HIPK4 showed the highest degree of this serine phosphorylation three residues downstream of the critical pY and also the highest absolute kinase activity in our radioactive kinase measurements (53.000 cpm HIPK4 versus 25.000 cpm HIPK3). This effect could possibly result from the second phosphorylation. Such additional phosphorylation in close proximity to the ATP-binding site could also affect the potency of the inhibitor abemaciclib. This is now discussed shortly in the revised version of the manuscript.

Lastly, as suggested by the Reviewer, we double-checked the electron density of residues S357, T515 and S541, which we found to be phosphorylated in 12%, 38% and 36% of all peptides, respectively, in the *Sf9* cell expressed protein used for crystallization (**last comment**). However, none of these side chains showed an extension of electron density at the side chain hydroxyl group that would indicate possible phosphorylation. This is coming back to the very first comment: As crystallization is considered a selection process for homogeneity, the fractions of heterogeneously phosphorylated protein samples might be discarded from the crystallization process.

• This reviewer suggests to reorganize the manuscript to better reflect the title. For example, moving apo-HIPK3 structure to a later part, so that it goes hand-in-hand with the inhibitor bound structures. In the discussion of crystal packing, please include its potential influence to compound binding interactions.

Following this as well as the first Reviewers' suggestion, we changed the title to "Abemaciclib is a potent inhibitor of DYRK1A and HIP kinases involved in transcriptional regulation". Since phosphorylation site and kinase activity analyses are now so tightly interconnected with structure (Figure 2, Supplementary Figure 3) we felt that changing the order would disrupt the flow of this study and would also not allow us to show the phosphorylation sites on the structure. In addition, we discuss the potential influence of crystal packing to the compound binding interactions in the revised version of the manuscript.

• The data processing statistics for apo-HIPK3 apo looks rather poor at 2.50Å resolution. Please consider more stringent resolution cutoff.

Thank you for the comment and yes, we agree at this point data processing statistics. However, the resolution cutoff decision was based on two considerations: a) Since the data were anisotropic (this is mentioned in the text), we used StarAniso that does several things: mainly, it truncates the data ellipsoidally, that results in low completeness. b) Including weak high-resolution data may improve the model, but these reflections never make the model worse (Diederichs, K. & Karplus, P. A. Better models by discarding data? *Acta Cryst.* **D69**, 1215–1222 (2013) and Evans, P. R. & Murshudov, G. N. How good are my data and what is the

resolution? *Acta Cryst.* **D69**, 1204–1214 (2013)). We now have included these two references in the revised manuscript. Because of these reasons, we did not "cut earlier". In addition, we refined the structure at lower resolutions, which resulted in very minor changes in the statistics but no changes in the structure. Lastly, we would like to comment that the resolution cutoff strategy is a matter of discussion among crystallography community since long and also widely-discussed in the CCP4 bulletin board. Therefore, even though, there can be a long discussion, we would request the reviewer to see our point.

We thank all Reviewers for their kind assessment of our study.

Reviewers' Comments:

Reviewer #1:

Remarks to the Author:

In general the authors have done a good job at addressing my concerns and adding clarifying data.

In principle I support publication, but do note I am still a little wary of the title. The new title still links transcriptional regulation and abemaciclib, when by the authors admission they have not been able to test this. I will defer to the editors recommendation on that.

Reviewer #2:

Remarks to the Author:

The authors have addressed all my comments in their revised manuscript in a thoughtful and comprehensive way. The additional experimental evidence requested is now included and enhances the contribution that this study will make to our understanding of kinase structure and mechanism.

Reviewer #3:

Remarks to the Author:

I thank the authors for the extensive revision. I have no further comments. The revised manuscript is suitable for publication.